# GENERATIVE MODELING WITH EXPLICIT MEMORY

## ABSTRACT

Conditional diffusion models require external guidance for generation, but common signals like text prompts are often noisy, necessitating prolonged training on massive, high-quality paired datasets. To address this, we introduce **G**enerative **M**odeling with **E**xplicit **M**emory (GMem), a framework that instead conditions generation on high-quality semantic information extracted directly from the data themselves. Such conditioning is stored in an external memory bank, providing an accurate guidance signal that can accelerate training by a large margin. Our experiments on ImageNet $256 \times 256$ show that GMem achieves a $50\times$ training speedup over SiT while also reaching a state-of-the-art (SoTA) FID of $1.53$. The key contributions of our work are threefold: (i) We demonstrate significant training acceleration on ImageNet datasets. (ii) We propose an efficient downstream adaptation pathway, where the image-pretrained model serves as a base model for adapting to new tasks. (iii) We introduce a data- and compute-efficient text-to-image (T2I) pipeline that matches the quality of strong baselines like PixArt-$\alpha$ using only $^1/_{17}$ of the data and $^1/_9$ of the training time. Our work establishes conditioning with explicit memory as a powerful paradigm for efficient and effective generative modeling. Our code will be made publicly available.

## 1 INTRODUCTION

Deep generative models like diffusion models (Yang & Wang, 2023; Ho et al., 2020; Song et al., 2020a; Nichol & Dhariwal, 2021; Choi et al., 2021; Li et al., 2024b; Chen et al., 2024b; Li et al., 2023a) have achieved notable success within the deep learning community. These methods demonstrate exceptional performance in complex tasks such as zero-shot text-to-image (T2I) and video generation (Podell et al., 2023; Saharia et al., 2022; Esser et al., 2024; Polyak et al., 2024; Brooks et al., 2024).

As data grow richer and model sizes become larger, training and sampling of diffusion models suffer from high computational burden (Karras et al., 2022; 2024). Moreover, the rapid growth of image data further amplifies the difficulty of obtaining large-scale, high-quality text prompts, exacerbating the cost of T2I training. Gu et al. (2023) show that due to memory capacity, scaling diffusion models on larger datasets requires proportionally more parameters, which is an increasingly unsustainable approach. This presents three key challenges to diffusion modeling: (i) faster training, (ii) reduced model memorization to facilitate easier adaptation to shifted domains, (iii) mitigating the reliance on text supervision during T2I training.

To address these interconnected challenges, we introduce GMem, a novel paradigm that synergizes representation learning with an external memory bank. Our approach is predicated on the observation that conditional generation tasks, such as T2I synthesis, inherently involve leveraging sparse guidance signals to generate dense, high-information content. We contend that in conventional T2I models, these guidance signals—typically text prompts—are susceptible to inaccuracies and noise. This paper investigates the conjecture that sparse yet salient information extracted directly from the images themselves can offer a more robust, efficient, and accurate form of guidance for the conditional generation process. To realize this, our GMem paradigm incorporates an external memory bank alongside the neural network, storing semantic representations to guide generation. This yields three key benefits: **(i) faster training**—the rich semantic information in memory bank accelerates convergence by reducing the burden of memorization; **(ii) easier adaptation**—the model supports both efficient domain adaptation to styles like anime or medical imaging via lightweight fine-tuning, and real-time user-guided editing through test-time memory manipulation. **(iii) data-efficient T2I**—our approach enables a text-to-image training pipeline that significantly reduces the reliance on massive paired datasets and prolonged training times. We also carefully analyze the relationship between

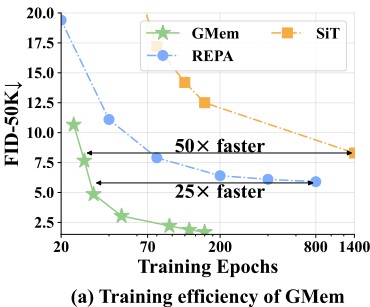 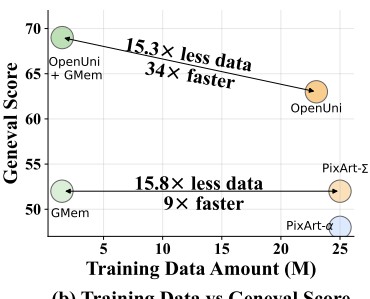

(a) Training efficiency of GMem  (b) Training Data vs Geneval Score

Figure 1: **GMem: appending an explicit memorization module into diffusion transformers unlocks data- and compute- efficiency gains** on ImageNet $256 \times 256$ and text-to-image generation. **Sub-figure (a)** demonstrates the training efficiency of GMem on ImageNet $256 \times 256$. At an FID $= 4.86$, GMem achieves over $25\times$ speedup compared to REPA (Yu et al., 2024). At an FID $= 7.66$, it achieves over $50\times$ speedup relative to SiT (Ma et al., 2024). **Sub-figure (b)** illustrates data- and compute- efficiency in text-to-image generation. Under comparable Geneval value, GMem requires $15.3\times$ fewer data and $34\times$ fewer training time comparing to OpenUni (Wu et al., 2025) baselines.

GMem and traditional diffusion models (e.g. DiT (Peebles & Xie, 2023)), and in Appendix E we present experiments demonstrating GMem can be progressively degraded into a standard conditional diffusion model, and vice versa.

We outline our contributions below:

(i) **Memory-augmented diffusion framework.** We introduce GMem, a memory-augmented framework for diffusion modeling that achieves $50\times$ training speedup on ImageNet $256 \times 256$, reaching FID=1.53 in only $\sim$20 hours training time. We also validate its effectiveness across various diffusion backbones and tokenizers.

(ii) **Rapid downstream adaptation.** An ImageNet-pretrained GMem can adapt to new domains (e.g., anime images, human faces) with only $\sim$20K fine-tuning steps (approximately 2 hours), achieving performance comparable to training from scratch.

(iii) **Data- and compute-efficient T2I training recipe.** We propose a bank-free T2I training pipeline that matches strong baselines like PixArt-$\alpha$ using only $^1/_{17}$ of the data (1.48M total images) and $^1/_9$ of the training cost (78.5 A100-days). This efficiency pattern extends to unified multimodal understanding and generation models and high-resolution text-to-image generation.

## 2  RELATED WORK

**Generative models.** Generative models——including Generative Adversarial Networks (GANs) (Goodfellow et al., 2014; Sauer et al., 2022; Xiao et al., 2021), Variational Autoencoders (VAEs) (Kingma, 2013; He et al., 2022), flow-based methods, and diffusion-based methods (Ho et al., 2020; Dhariwal & Nichol, 2021; Mittal et al., 2023)——aim to learn the data distribution $p(\mathbf{x})$ and generate data through sampling, achieving remarkable performance in producing realistic samples (Li et al., 2023b). Recently, diffusion-based methods employ stochastic interpolation to model a forward process and then reverse the Gaussian distribution back to the original image space, generating realistic samples. These methods achieve SoTA results in deep generative modeling and are the focus of this study (Mittal et al., 2023; Song & Ermon, 2020; Durkan & Song, 2021).

Diffusion models face computational challenges due to high training cost and instability (Yu et al., 2024; Song & Ermon, 2020) and high sampling costs from multi-step generation (Lu & Song, 2024), driving extensive research to accelerate both processes. For example, REPA (Yu et al., 2024) leverages external visual representations to speed up training. LightningDiT (Yao & Wang, 2025) accelerates training by aligning the latent space of vision tokenizer (i.e. VA-VAE) with pretrained vision encoder. Instead, GMem constructs an explicit memory bank of semantic representations to guide the model toward richer feature learning, thereby significantly accelerating training and sampling.

**Diffusion modeling and representation learning.** To overcome the instability and computational inefficiency of diffusion models, recent studies (Yu et al., 2024; Fuest et al., 2024; Mittal et al., 2023) start to leverage representation learning to enhance diffusion models. On the one hand, diffusion models are capable of learning high-quality representations (Yu et al., 2024). For instance, Tang et al. (2023) demonstrate that feature maps extracted from diffusion networks can establish correspondences between real images, indicating a strong alignment between the learned representations with actual

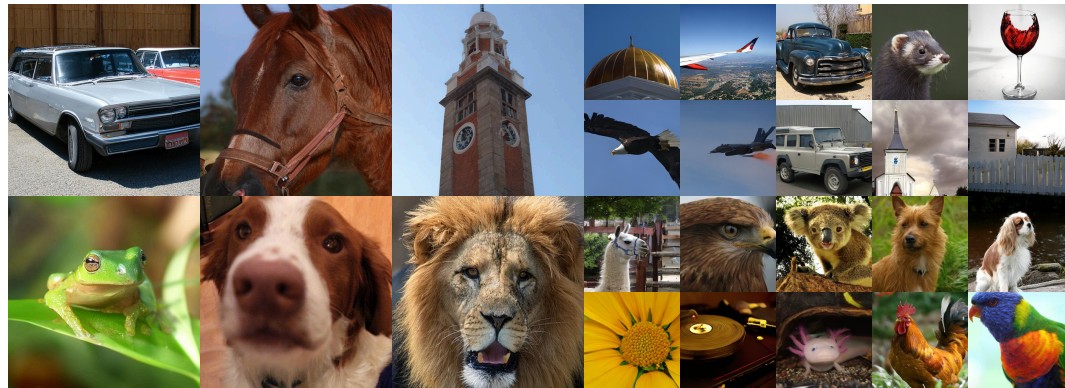

Figure 2: **Selected samples on ImageNet $512 \times 512$ and $256 \times 256$.** This figure presents images generated by GMem under two experimental settings: (i) For ImageNet $256 \times 256$, GMem was trained for 160 epochs and sampled via Euler method (NFE $= 100$), achieving an FID $= 1.53$ without classifier-free guidance. (ii) For ImageNet $512 \times 512$, training extended to 400 epochs with identical sampling settings, yielding FID $= 1.89$.

image. Furthermore, Yang & Wang (2023) conduct a detailed analysis of the trade-off between the quality of learned representations and the penalization of the optimal parameter spectrum.

On the other hand, well-trained representation models can improve performance and expedite the training of diffusion models. Mittal et al. (2023) accomplish this by adjusting the weighting function in the denoising score matching objective to enhance representation learning. REPA (Yu et al., 2024) introduces an alignment loss for intermediate layer representations, significantly accelerating the training process by over $17.5$ times.

For a detailed discussion of retrieval-augmented generation (RAG) methods and other related approaches, please refer to Appendix B

## 3 MOTIVATION

### 3.1 THE CONTINUUM OF GENERATIVE PARADIGMS AND GMEM

**Conditioning strength controls velocity complexity.** The difficulty of training a flow-matching model is primarily determined by the geometric complexity of its velocity field $\mathbf{v}_\theta(\mathbf{x}_t, t, \mathbf{c})$. We argue this complexity, in turn, depends on the *conditioning strength* of $\mathbf{c}$: stronger, more informative conditioning tends to simplify the vector field, while weaker conditioning leads to more curved and entangled trajectories. In the ideal case where $\mathbf{x}_0$ is fully specified (i.e., the conditioning directly pins down the target), the probability flow can be represented by (almost) straight-line trajectories from the noise prior to $\mathbf{x}_0$, yielding an approximately linear velocity field and hence the easiest optimization.

**Three paradigms of generative modeling.** Based on this perspective, generative models are categorized into distinct tasks based on the density of $\mathbf{c}$: (i) **Unconditional generation** ($\mathbf{c} = \emptyset$) operates with zero guidance, mapping noise to data without directional cues; (ii) **Class-conditional generation** relies on discrete labels ($\mathbf{c} \in \{1, \ldots, K\}$), providing sparse, cluster-level constraints; (iii) **Text-to-Image (T2I) generation** utilizes dense semantic embeddings ($\mathbf{c} \in \mathbb{R}^d$), offering fine-grained, high-density supervision. Despite sharing the same diffusion formalism, these paradigms are traditionally treated as a separate yet progressive sequence of tracks.

**GMem as a unified bridge.** Building on this view of conditioning density, GMem leverages a scalable memory bank to continuously tune the information content of $\mathbf{c}$, spanning from weak class-level guidance to dense, instance-level cues. Within a single GMem model, simply manipulating the memory bank (e.g., from class snippet to instance-specific snippets) enables seamless conversion between GMem and traditional class conditional method as described in Section 5.2 . By attaching a lightweight T2S adapter that maps text embeddings into the snippet space, we convert dependence of GMem on snippet-based inputs into text-based conditioning and fine-tune the pretrained GMem backbone into an end-to-end T2I generator Section 4.4 . Our experiments ( Section 5.4 and Appendix E ) empirically validate these conversions between class-conditional and T2I configurations on top of a pretrained GMem backbone, providing strong evidence that GMem forms a data- and compute-efficient bridge between these generative paradigms.

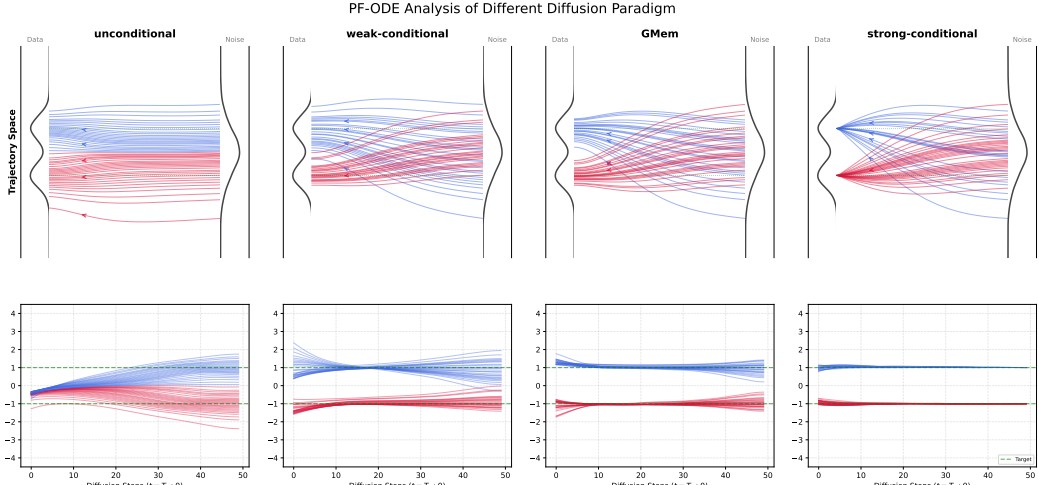

Figure 3: **Analysis of conditioning strength on learning dynamics. (Top Row) Probability flow trajectories:** GMem effectively rectifies the flow trajectories similarly to the strong-conditional baseline, minimizing curvature and trajectory length, which empirically correlates with easier training. Crucially, GMem preserves the distributional spread seen in weak conditioning. **(Bottom Row) Evolution of predicted $x_0$:** The rapid convergence to the target and the subsequent flatness of the curves indicate a linearized velocity field, confirming that GMem simplifies the learning landscape compared to the oscillating predictions of weak baselines.

## 3.2 GUIDANCE STRENGTH AND FLOW CURVATURE

To make this perspective concrete, we analyze the learning dynamics of a parameterized velocity field $v_\theta(x_t, t, c)$ on a Gaussian Mixture Model under four conditioning regimes of increasing strength:

(i) **Unconditional generation ($c = \emptyset$).** The model receives no external guidance and learns to approximate the marginal data distribution $p(x_0)$.

(ii) **Weak conditional generation ($c = \mu_k$).** Conditioning on the cluster mean $\mu_k$. This mirrors standard class-conditional generation, providing global structural guidance.

(iii) **GMem generation ($c = s_{k^*} + \epsilon$, $k^* = \arg\min_k \|x_0 - s_k\|_2$).** We subsample memory snippets $\{s_k\}$ from the training data and, for each $x_0$, define $c = s_{k^*} + \epsilon$ where $\epsilon \sim \mathcal{N}(0,1)$. This provides finer but still lossy guidance, interpolating between class labels in (i) and full observations in (iv), in line with Section 4 .

(iv) **Strong conditional generation ($c = x_0$).** The "oracle" setting where the exact target coordinate is provided. This theoretically implies a deterministic mapping and represents the limit of high-density conditioning, such as text-to-image models with highly detailed prompts.

We visualize the resulting velocity fields and probability flow trajectories from our numerical simulations in Figure 3 .

**Weak labels induce high-curvature flows.** In unconditional and weak-conditional regimes, the conditional information is insufficient to localize the target $x_0$ at high noise levels ($t \to T$). Mathematically, the optimal prediction collapses to the conditional expectation of the distribution, which corresponds to the geometric centroid: $\lim_{t \to T} \mathbb{E}[x_0 \mid x_t, c] \approx \mathbb{E}[x_0 \mid c]$. Consequently, early trajectories regress toward this global mean. However, as $t \to 0$, the flow is compelled to bifurcate sharply to fit the multi-modal data distribution. This misalignment between the initial mean-seeking dynamics and the final mode-seeking requirement forces the velocity field $v(x, t)$ to exhibit extreme non-linearity and curvature, significantly impeding convergence.

**Strong label suffers from overfitting.** Providing the exact target $c = x_0$ resolves ambiguity, effectively reducing the conditional entropy to zero: $H(x_0 \mid x_t, c) \to 0$. While this straightens the flow trajectories and simplifies the velocity matching objective, it forces the generative distribution to collapse into a Dirac delta function: $p_\theta(x \mid c) \to \delta(x - c)$. This results in severe overfitting, where the model essentially predict the same data under the same condition across different input noise.

**GMem balances flow straightening and overfitting.** GMem functions as a conceptual bridge connecting the weak and strong conditioning paradigms. By modulating the granularity of $c$, GMem

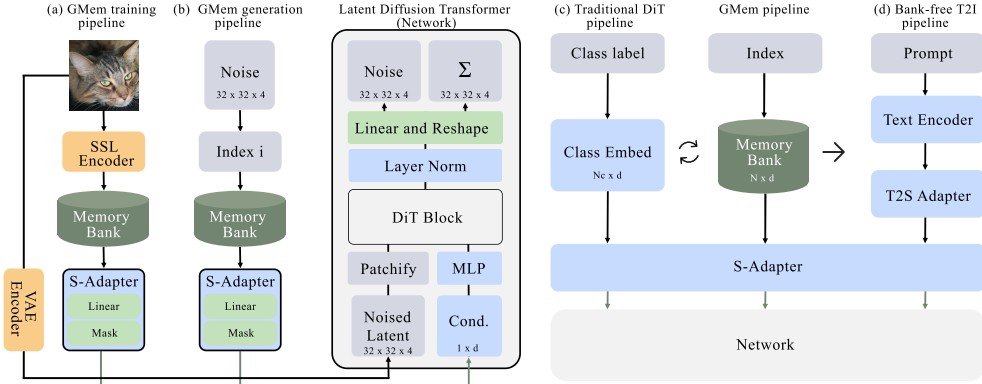

Figure 4: (Left) **Training and inference pipeline with GMem. Sub-figure (a): Training pipeline of GMem.** During training, network is conditioned on processed snippets after S-Adapter. **Sub-figure (b): Sampling pipeline of GMem.** At inference time, given a random noise $\epsilon$, a memory snippet is retrieved from the memory bank and fed into the network after S-Adapter. (Right) **Comparison of GMem pipeline with other generative pipelines. Sub-figure (c): Degrads GMem to a standard diffusion model.** By replacing the memory bank with a class–conditioned embedding table (one snippet per label), GMem degrades to a standard diffusion model. **Sub-figure (d): Upgrads GMem to T2I generation.** By replacing the memory bank with a pretrained text encoder and a lightweight Text-to-snippet (T2S) adapter, GMem upgrades to a text-to-image generation model. On the right, the arrows indicate that these frameworks are unified by GMem and can be converted into one another by simply swapping the conditioning module.

effectively interpolates the conditional entropy: $H_{\text{strong}} < H_{\text{GMem}} < H_{\text{weak}}$. Empirically, this design strikes a balance: (i) unlike weak labels, GMem provides sufficiently dense semantic anchors to rectify the flow curvature, accelerating training; (ii) unlike strong labels, the injected noise $\epsilon'$ maintains necessary uncertainty, preventing the distribution from collapsing into a Dirac delta.

## 4 METHODOLOGY

Current generative models (Ho et al., 2020; Song & Ermon, 2020) typically rely on one neural network to simultaneously achieve both generalization and memorization of data distributions. However, the capacity-constrained modern diffusion models, such as UNet and Transformer-based networks (Ho et al., 2020; Peebles & Xie, 2023), face two critical limitations (Kadkhodaie et al., 2023): (i) insufficient model parameters to memorize complex data distributions, and (ii) computationally expensive parameter optimization for memorization.

To address these limitations, we propose augmenting the architecture with an external memory bank—an editable memory mechanism that enables fast adaptation to other unseen domains. The proposed framework is illustrated in Figure 4.

### 4.1 EXTERNAL MEMORY BANK CONSTRUCTION

Building on Kadkhodaie et al. (2023)'s insight that diffusion models achieve generalization through geometry-adaptive harmonic representations, we design a memory bank for diffusion models to: (i) supply essential semantic information for generating high-quality, realistic images; and (ii) exclude excessive details to prevent overfitting to the training data while maintaining robust generalization.

**Memory snippet extraction.** We employ a representation model $f$ to extract semantic features from any input $\mathbf{x} \sim D$. We refer to $f(\mathbf{x}) \in \mathbb{R}^d$ as the global image representation produced by $f$ (e.g., the last-layer [CLS] representation in DINOv2), where $d$ is the dimensionality of the representation. Finally, we define the memory snippet $\mathbf{s} \in \mathbb{R}^d$ as the $\ell_2$-normalized global image representation: $\mathbf{s} = f(\mathbf{x}) / \|f(\mathbf{x})\|_2$.

**Memory bank construction.** We then construct the memory bank by collecting $N$ such snippets from the training dataset $D$. Formally, the memory bank is represented as a matrix $\mathbf{M} \in \mathbb{R}^{N \times d}$, composed of $N$ unit-norm snippets: $\mathbf{M} = [\mathbf{s}_1, \mathbf{s}_2, \ldots, \mathbf{s}_N]^\top$ where $|\mathbf{s}_i|_2 = 1$.

**Representation models for memory bank construction.** GMem supports diverse representation models $\boldsymbol{f}$, each varying in their ability to capture image information, which also directly influences generative model performance. We employ self-supervised representation models for two key reasons:

(i) Yu et al. (2024) demonstrate that self-supervised models (e.g. DINOv2 (Oquab et al., 2023)) can expedite the training of diffusion models. It motivates to use it in GMem.

(ii) Self-supervised models capture semantic information more effectively than supervised alternatives (Bordes et al., 2022; Zimmermann et al., 2021; Sun et al., 2024).

We also examine other representation models, including the CLIP visual encoder (Radford et al., 2021), to construct the memory bank (see Appendix G.5).

**Incorporating memory bank into diffusion models.** As illustrated in Figure 4 (right), the key architectural difference between GMem and traditional diffusion transformers (e.g., DiT (Peebles & Xie, 2023)) lies in the use of memory snippets as conditioning information.

To be specific, we introduce a **Snippet Adapter (S-Adapter)** that transforms raw memory snippets into the model's conditioning space through a two-stage process:

(i) *Linear*: The normalized snippet is projected to match the diffusion transformer's hidden dimensionality $d_t$: $\mathbf{s}_{\mathrm{proj}} = \mathbf{W}_{\mathrm{proj}}\tilde{\mathbf{s}} + \mathbf{b}_{\mathrm{proj}}$, where $\mathbf{W}_{\mathrm{proj}} \in \mathbb{R}^{d_t \times d}$ and $\mathbf{b}_{\mathrm{proj}} \in \mathbb{R}^{d_t}$ are learnable.

(ii) *Mask*: To prevent overfitting to specific snippets, we apply random feature-level masking during training: $\mathbf{s}_{\mathrm{masked}} = \mathrm{MASK}(\mathbf{s}_{\mathrm{proj}})$. We tested three masking designs in Section G.5 and use the *random masking* strategy, where each dimension of $\mathbf{s}_{\mathrm{proj}}$ has a probability $p$ of being set to zero.

Consistent with the design of class embeddings in DiT, the processed snippet $\mathbf{s}_{\mathrm{masked}}$ is added to the timestep embedding $\mathbf{e}_t$ to form the final conditioning signal: $\mathbf{c} = \mathbf{e}_t + \mathbf{s}_{\mathrm{masked}}$.

**Bridging GMem and DiT.** We argue that the memory snippets $\mathbf{s}$ serve as a more general conditioning signal compared to learned class embeddings, facilitating cross-dataset adaptation. More importantly, as illustrated in Figure 4 (c), GMem can be systematically converted to a standard DiT by replacing the memory bank with class embeddings, and vice versa. This dual conversion reveals that DiT is essentially a degenerate case of GMem with a highly compressed memory bank. We further provide comprehensive experimental validation of this bidirectional conversion in Appendix E.

## 4.2 TRAINING WITH MEMORY BANK

Memory bank provides semantic information about the data distribution, aiding both training and inference phases in diffusion models. To further integrate snippet into training, we adapt the training loss of diffusion models as:

$$\mathcal{L}(\boldsymbol{\theta}) = \int_0^T \mathbb{E}\|\mathbf{v}_{\boldsymbol{\theta}}(\mathbf{x}_t, \mathbf{s}, t) - \dot{\alpha}_t\mathbf{x}_0 - \dot{\sigma}_t\boldsymbol{\epsilon}\|^2 dt\,, \tag{1}$$

where $\mathbf{x}_0 \sim D$, $\mathbf{s} = f(\mathbf{x}_0)/\|f(\mathbf{x}_0)\|_2$, $\boldsymbol{\epsilon} \sim \mathcal{N}(0, \mathbf{I})$, $\dot{\alpha}_t = \frac{\mathrm{d}\alpha_t}{\mathrm{d}t}$, $\dot{\sigma}_t = \frac{\mathrm{d}\sigma_t}{\mathrm{d}t}$, and $\mathbf{v}_{\boldsymbol{\theta}}$ is the velocity estimated by the network.

## 4.3 SAMPLING WITH MEMORY BANK

During training, we store the memory bank for use at generation. The indexing-based retrieval method during sampling introduces minimal additional overhead. We also design an efficient storage scheme to reduce the storage cost. Additionally, the editability of the memory bank enables the model to perform test-time domain adaptation.

**Sampling pipeline.** The generation pipeline using memory bank works as (see Figure 4 (b)):

(i) Generation begins by sampling input noise $\mathbf{z} \sim \mathcal{N}(0, \mathbf{I})$. The size of $\mathbf{z}$ typically matches that of the VAE latent (e.g. $4 \times 32 \times 32$ for SD-VAE (Podell et al., 2023)), and acts as $\mathbf{x}_T$, a heavily noised image awaiting denoising, where $T$ is the total number of diffusion steps.

(ii) To enable end-to-end generation (i.e., relying solely on the input noise $\mathbf{z}$), we devised a noise-based indexing mechanism: mapping the input noise $\mathbf{z}$ to a uniformly distributed index $i = \Phi(\mathbf{z})$, where $\Phi(z) = \frac{1}{\sqrt{2\pi}}\int_{-\infty}^z e^{-\frac{1}{2}u^2}du$ is the standard normal cumulative distribution function.

(iii) Using the index $i$, we retrieve the corresponding memory snippet $\mathbf{s}_i$ from the memory bank.

(iv) Finally, we feed the noise $\mathbf{z}$ and the retrieved snippet $\mathbf{s}_i$ into the network to iteratively refine the noise into a high-quality image.

Table 1: **Sampling quality on various datasets.** We report the performance of GMem on CIFAR-10 (left), ImageNet $256 \times 256$ (middle) and ImageNet $512 \times 512$ (right). GMem achieves comparable FID with fewer training epochs across multiple datasets. All results reported are w/o classifier-free guidance unless otherwise specified.

| CIFAR-10 | | | ImageNet $256\times256$ | | | ImageNet $512\times512$ | | |
|---|---|---|---|---|---|---|---|---|
| **METHOD** | **Epoch** ($\downarrow$) | **FID** ($\downarrow$) | **METHOD** | **Epoch** ($\downarrow$) | **FID** ($\downarrow$) | **METHOD** | **Epoch** ($\downarrow$) | **FID** ($\downarrow$) |
| **Traditional generative models** | | | **Traditional generative models** | | | **Traditional generative models** | | |
| BigGAN (Brock, 2018) | - | 14.7 | BigGAN (Brock, 2018) | - | 6.96 | StyleGAN-XL (Sauer et al., 2022) | - | 2.41 |
| StyleGAN2 (Karras et al., 2020) | - | 8.32 | VQ-GAN (Esser et al., 2021) | - | 15.78 | BigGAN (Brock, 2018) | 472 | 9.54 |
| **Diffusion models (UNets)** | | | **Diffusion models (UNets)** | | | **Diffusion models (UNets)** | | |
| DDPM (Ho et al., 2020) | 2048 | 3.17 | ADM (Dhariwal & Nichol, 2021) | 400 | 10.94 | ADM (Karras et al., 2024) | - | 23.2 |
| DDIM (Song et al., 2020a) | - | 4.04 | | | | EDM2 (Karras et al., 2024) | 734 | 1.91 |
| Score SDE (deep) (Song et al., 2020b) | - | 2.20 | **Diffusion models (Transformer)** | | | **Diffusion models (Transformer)** | | |
| EDM (Karras et al., 2022) | 4000 | 2.01 | MaskGIT (Chang et al., 2022) | 300 | 6.18 | MaskGIT (Chang et al., 2022) | 300 | 7.32 |
| Diffusion Style-GAN (Wang et al., 2022a) | - | 3.19 | MAGVIT-v2 (Yu et al., 2023) | 270 | 3.65 | MAGVIT-v2 (Yu et al., 2023) | 270 | 3.07 |
| Diffusion GAN (Xiao et al., 2021) | 1024 | 3.75 | SD-DiT (Zhu et al., 2024) | 480 | 7.21 | DiT-XL/2 (Peebles & Xie, 2023) | 600 | 12.03 |
| **Diffusion models (Transformer)** | | | DiT-XL/2 (Peebles & Xie, 2023) | 1400 | 9.62 | SiT-XL/2 (Ma et al., 2024) (w/ cfg) | 600 | 2.62 |
| SiT-XL/2 (Ma et al., 2024) | 512 | 6.68 | SiT-XL/2 (Ma et al., 2024) | 1400 | 8.30 | + REPA (Yu et al., 2024) (w/ cfg) | 200 | 2.08 |
| + REPA (Yu et al., 2024) | 200 | 4.52 | + REPA (Yu et al., 2024) | 782 | 5.90 | LightningDiT-XL/1 (Yao & Wang, 2025) | - | - |
| + GMem (ours) | 52 | **4.08** | + GMem (ours) | 80 | 5.27 | + GMem (ours) | 400 | **1.89** |
| + GMem (ours) | 450 | **1.22** | LightningDiT-XL/1 (Yao & Wang, 2025) | 800 | 2.17 | + GMem (ours) (w/ cfg) | 400 | **1.71** |
| | | | + REPA (Yu et al., 2024) | 160 | **1.84** | | | |
| | | | + GMem (ours) | 160 | **1.53** | | | |

For privacy concerns regarding memory banks derived from training data, we retain banks for small datasets (e.g., ImageNet) with leakage analysis in Appendix G.1 , while replacing banks with lightweight text-to-snippet adapters for large-scale T2I settings ( Section 4.4 ).

**Memory compression and manipulation.** Large-scale memory banks can be compressed using SVD decomposition, reducing storage from $\mathcal{O}(Nd)$ to $\mathcal{O}(Nr+dr)$ where $r \ll \min(N, d)$ (details in Appendix C ). Additionally, the memory bank supports test-time adaptation through external snippet incorporation or interpolation between existing snippets, enabling novel style generation without retraining; implementation details are provided in Appendix D.1 .

## 4.4 TEXT-TO-IMAGE GENERATION

To bypass the prohibitive storage cost of an explicit memory bank in large-scale T2I, we propose a **bank-free pipeline** shown in Figure 4 (d). The core idea is to first map a text prompt to a snippet using a lightweight **Text-to-Snippet (T2S) module**; this snippet is then processed by the pretrained **S-Adapter** and fed into the pretrained GMem network to synthesize the final image.

**T2S Module Design.** The T2S module consists of two sequential components:

(i) *Text Encoder*: A frozen pretrained encoder (e.g., Gemma-2B (Team et al., 2024)) that transforms input text into intermediate latent representations $\mathbf{T}$.

(ii) *T2S Adapter*: A lightweight two-layer multilayer perceptron (MLP) that maps text representations to the snippet space: $\mathbf{s} = g_\phi(\frac{1}{L} \sum_{i=1}^{L} \mathbf{T}_i)$, where $g_\phi$ is the MLP.

The generated snippet $\mathbf{s}$ is then processed by the pretrained **S-Adapter** and fed into pretrained GMem network.

**Advantages of the Bank-Free T2I Design.** Replacing the explicit memory bank with a learnable T2S module offers three key advantages over traditional T2I approaches. *(i) Network reusability and training efficiency:* by decoupling text-to-snippet mapping from snippet-to-image synthesis, we can directly reuse pretrained GMem networks without retraining from scratch. *(ii) Privacy preservation relative to retrieval-based methods:* instead of storing per-example features in an external memory bank, our approach internalizes semantic information into model parameters, avoiding the additional privacy risks associated with persistent retrieval databases (e.g., RDM (Blattmann et al., 2022)). *(iii) Storage efficiency relative to retrieval-based methods:* the T2S module replaces the potentially massive retrieval memory banks that may contain millions of snippets or feature vectors. We argue that this reduction in extra storage overhead beyond the base model is especially important in T2I tasks that typically involve large-scale datasets.

## 5 EXPERIMENTS

### 5.1 EXPERIMENTAL SETTING

**Datasets.** For pixel-level generation, we test GMem on CIFAR-10 (Krizhevsky et al., 2009) due to its diverse classes and its popularity in benchmarking image generation. We then evaluate GMem on ImageNet $256 \times 256$ (Deng et al., 2009) to examine how it models latent space distributions, which is a key focus in recent image generation research (Karras et al., 2022; Yu et al., 2024; Ma et al., 2024; Peebles & Xie, 2023). Finally, we assess the scalability of GMem to larger resolutions by conducting experiments on ImageNet $512 \times 512$ (Deng et al., 2009).

**Backbones and visual tokenizers.** Following prior image generation approaches (Yu et al., 2024), we primarily use LightningDiT (Yao & Wang, 2025) and SiT (Ma et al., 2024) as the backbone. We also evaluate the effectiveness of GMem on different visual tokenizers such as SD-VAE (**?**), DC-AE (Chen et al., 2024a), and VA-VAE (Yao & Wang, 2025) in Table 12 .

**Baselines.** Unless noted otherwise we report numbers for two SoTA, efficiency-oriented backbones. *(i) SiT (Ma et al., 2024)+REPA (Yu et al., 2024):* one of the strongest and most efficient diffusion models. *(ii) LightningDiT (Yao & Wang, 2025):* a highly optimised diffusion transformer with competitive FID and fast convergence. All speed (training epochs, NFEs, wall-clock sampling time) and quality (FID) results are shown before and after inserting GMem, isolating the contribution of GMem. For overall image-generation quality we additionally list the best reported numbers, as shown in Table 1 , from the latest diffusion models, additional baselines are provided in Appendix F.3 .

**Metrics.** For image generation, consistent with prior work (Peebles & Xie, 2023; Yu et al., 2024), we primarily use FID-50K (FID) (Heusel et al., 2017) to evaluate generation quality. For T2I tasks, aligned with (Chen et al., 2023), we primarily use GenEval (Ghosh et al., 2023) as the evaluation metric, supplemented by MJHQ-50K (Li et al., 2024a). For efficiency metrics, we primarily use A100 GPU-days (GPUd) to measure training or sampling time overhead. We apply an exchange rate of 1/2.4 to convert H800 GPU days to A100 equivalents. Consistent with prior work (Borgeaud et al., 2022; Wu et al., 2022), we count only parameters updated by gradients during training.

**Experimental details.** Unless stated otherwise we keep the original training pipelines of each backbone (Ma et al., 2024; Yu et al., 2024; Yao & Wang, 2025). We list the pipelines below: *(i) data preprocessing:* exactly as in the backbone papers. For data augmentation, SiT uses the raw images with no augmentation; LightningDiT applies only random horizontal flips. For latent models we adopt SD-VAE (**?**) with SiT and VA-VAE (Yao & Wang, 2025) (f16d32, patch size $= 1$) with LightningDiT. *(ii) model configuration:* unless noted we report results for SiT-XL/2+REPA and LightningDiT-XL/1. Smaller settings SiT-L/2+REPA and LightningDiT-B/1 are also used in ablations. *(iii) training strategy:* AdamW with batch size 256 for SiT and 1024 for LightningDiT, matching (Song et al., 2020b; Karras et al., 2024). *(iv) sampler configurations:* following SiT (Ma et al., 2024), we use SDE solver and set the NFE to 50 for CIFAR-10 and 100 for ImageNet $256 \times 256$ and $512 \times 512$ by default. Full implementation details appear in Appendix F .

## 5.2 Enhanced Generation with Memory Bank

As illustrated in Section 4.2 and Figure 4 , we train the network with memory bank and study the benefits brought by GMem. Implementation details for each experiment are provided in Appendix F .

**Training efficiency.** A key advantage of GMem lies in its ability to substantially improve training efficiency. We evaluate how rapidly GMem attains target image quality compared to diffusion-transformer baselines under two generative regimes.

 (i) *pixel-space generation:* On CIFAR-10 ( Table 1 ), GMem matches REPA's performance within 52 epochs, yielding a $3.85\times$ speedup over REPA and over $10\times$ compared to SiT.

 (ii) *latent-space generation:* On ImageNet ( Table 1 ), GMem reaches FID$=1.53$ at $256\times256$ and FID$=1.71$ at $512\times512$ in only 160 and 400 epochs, respectively, without classifier-free guidance (CFG). As summarized in Table 9 , GMem attains competitive FID with much fewer epochs than the most efficient baselines: (i) Using only 32 epochs ($\mathbf{25\times}$ speedup), GMem achieves FID$=4.86$, outperforming REPA which requires $800+$ epochs; (ii) With 28 epochs ($\mathbf{50\times}$ speedup), GMem delivers FID$=7.66$, surpassing SiT's FID$=8.61$ at 1400 epochs.

Furthermore, through a comparison between generating and training images (see Appendix G.2 ), we confirm that these efficiency gains are not achieved at the expense of generative diversity.

**Other benefits.** Due to space limit, we highlight several additional benefits of GMem in the appendix.

 (i) *Sampling efficiency:* in Appendix G.3 , we show that in addition to training efficiency, GMem delivers up to a $5\times$ **sampling speedup** on ImageNet $256 \times 256$.

 (ii) *Memory manipulation enables test-time editing:* in Appendix D.1 , we show that GMem can synthesize novel concept compositions (e.g., "a dog wearing a hat") by manipulating of snippet during test-time. We argue such controllable interpolation underscores potential of GMem for real-time, user-guided image editing.

Table 2: **Efficiency advantages of GMem.** (Left) **Adaptation to specialized domains.** GMem achieves zero-shot performance superior to DiT-XL/2 on FFHQ and matches PixArt-$\alpha$ on MJHQ with only 20K fine-tuning steps. (Middle) **Training speedup on ImageNet** $256 \times 256$. GMem delivers up to $50\times$ training speedup while maintaining or improving generation quality on ImageNet $256 \times 256$. (Right) **Memory bank capacity.** Larger memory bank consistently improves the performance of GMem. All experiments are conducted using LightningDiT on ImageNet.

| Setting | Fine-tuning budget (steps) | FID ($\downarrow$) |
|---|---|---|
| *FFHQ (Karras et al., 2019)* | | |
| Pretrained GMem | 0 | 8.65 |
| Pretrained GMem | 20K | 6.62 |
| DiT-XL/2 (Peebles & Xie, 2023) | – | 12.86 |
| *MJHQ (Li et al., 2024a)* | | |
| Pretrained GMem | 0 | 8.73 |
| Pretrained GMem | 20K | 6.10 |
| PixArt-$\alpha$ (Chen et al., 2023) | – | 6.14 |

| Method | # Params | Epoch ($\downarrow$) | FID ($\downarrow$) |
|---|---|---|---|
| DiT-XL/2 | 675M | 1400 | 9.62 |
| SiT-XL/2 | 675M | 1400 | 8.61 |
| + REPA | 675M | 800 | 5.90 |
| LightningDiT-XL/1 | 675M | 800 | 2.17 |
| + GMem | 684M | 160 | 1.58 |
| + REPA | 675M | 160 | 1.84 |
| + GMem | 684M | 28 | 7.66 |
| + GMem | 684M | 32 | 4.86 |
| + GMem | 684M | 160 | 1.53 |

(iii) ***Dual conversion between GMem and DiT:*** in Appendix E, we demonstrate that GMem and DiT are inter-convertible. We show that a standard DiT can be viewed as a specialized instance of GMem with a highly compressed memory bank (i.e., class embeddings), and provide experimental validation for converting between both frameworks.

### 5.3 EFFICIENT DOWNSTREAM ADAPTATION

In this section, we validate GMem's adaptation capabilities and training efficiency on downstream tasks. We use a GMem model pretrained on ImageNet $256 \times 256$ for $650$ epochs (referred to as Pretrained GMem) as the base model for fine-tuning on downstream datasets.

**Adaptation to specialized domains.** We evaluate GMem's adaptation capabilities on FFHQ (Karras et al., 2019) (high-quality human faces) and MJHQ (Li et al., 2024a) (anime-style images)—detailed experimental settings are provided in Appendix H.2. Results are summarized in Table 2.

Main results demonstrate both zero-shot transfer and rapid fine-tuning capabilities of GMem:

(i) ***Zero-shot transfer:*** On FFHQ, Pretrained GMem achieves FID=8.65, outperforming DiT-XL/2 trained from scratch (FID=12.86). We argue that face images share semantic similarities with natural images, enabling effective knowledge transfer from ImageNet pretraining.

(ii) ***Rapid adaptation:*** On MJHQ, which differs significantly from ImageNet in visual style, 20K fine-tuning steps enable GMem to achieve FID=6.10, matching PixArt-$\alpha$. This demonstrates that even when pretraining provides limited transferable knowledge, GMem can efficiently adapt to new downstream tasks.

Additional experimental results on medical imaging domains are provided in Appendix H.2.

These results establish Pretrained GMem as **a general-purpose base model** that requires only minimal computational investment ($\sim$20K steps, approximately 2 hours on $8\times$H800 GPUs) to adapt to specialized domains. This efficiency is particularly valuable in scenarios with limited training data or computational resources, making **rapid downstream adaptation as a core advantage of GMem**.

**Adaptation to higher resolutions.** The efficient adaptation capability of GMem also holds when handling different resolutions: with just approximately $0.3$ epochs of fine-tuning, pretrained GMem is able to generate high-quality, high-resolution face images (FID =11.57 on MJHQ $1024 \times 1024$). The experimental results can be found in Appendix G.4.

### 5.4 DATA- AND COMPUTE-EFFICIENT TEXT-TO-IMAGE GENERATION

Following the pipeline in Section 4.4, we transform Pretrained GMem into an end-to-end T2I model by replacing the memory bank with a lightweight T2S adapter. We evaluate this pipeline across three dimensions: generation quality (GenEval (Ghosh et al., 2023)), data efficiency, and compute efficiency. Results are summarized in Table 15 with detailed experimental settings in Appendix H.3.

**Text-to-image generation.** We assess GMem's T2I capabilities against both memorization and diffusion baselines, with key findings listed below.

(i) ***Data efficiency.*** Compared to PixArt-$\alpha$ (Chen et al., 2023) which requires 25M private paired samples, GMem uses only 0.2M open-sourced paired data plus 1.28M ImageNet images for pretraining ($1/17$ reduction). This result demonstrats that expensive large-scale paired datasets are not necessary for high-quality T2I generation.

Table 3: **Validation of compute and data efficiency of GMem.** (Left): **Text-to-image generation.** (Right): **Unified multimodal understanding and generation.** GMem achieves competitive performance across various networks and resolutions using approximately $1/9$ of the training time and $1/17$ of the data compared to baselines.

| Model | GPUd | Data amount | MJHQ FID ($\downarrow$) | GenEval ($\uparrow$) |
|---|---|---|---|---|
| GMem | 75+3.5 | 1.28M + 0.20M | 6.32 | 0.52 |
| CLIP retrieval | – | 5B | – | 0.35 |
| PixArt-$\alpha$ | 753 | 25M | 6.14 | 0.48 |
| SD 1.5 | 6,250 | 2,000M | 9.62 | – |

| Model | Resolution | Pretraining epochs | Training epochs | GenEval ($\uparrow$) |
|---|---|---|---|---|
| OpenUni | 512 | 2.226 | - | 0.63 |
| + GMem | 512 | 0.056 | 0.009 | 0.69 |
| OpenUni | 1024 | 2.226 | - | 0.60 |
| + GMem | 1024 | 0.089 | 0.009 | 0.64 |

(ii) *Compute efficiency.* GMem achieves GenEval= 0.52 using only 78.5 GPU-days total training time, representing a $1/9$ reduction compared to PixArt-$\alpha$'s 753 GPU-days. We argue this efficiency stems from leveraging knowledge transfer from ImageNet pretraining, eliminating the need for extensive training T2I from scratch.

(iii) *Beyond Retrieval.* GMem attains a $\sim 48\%$ relative improvement over CLIP retrieval (GenEval=0.35), demonstrating its ability to synthesize novel, text-aligned images rather than merely retrieving existing ones.

**Unified multimodal understanding and generation.** We further evaluate whether the proposed T2I training recipe generalizes to unified multimodal understanding and generation scenarios (Zhang et al., 2025). Specifically, we adopt OpenUni (Wu et al., 2025) as the network backbone, with detailed experimental settings provided in Appendix H.1 . Results are summarized in Table 3 .

The findings demonstrate that GMem maintains its data and compute efficiency advantages:

(i) *Compute efficiency.* At both $512 \times 512$ and $1024 \times 1024$ resolutions, GMem achieves superior GenEval scores with less than $1/34$ and $1/23$ of the training epochs respectively, reinforcing its computational efficiency across different scales.

(ii) *Data efficiency.* GMem demonstrates significant data efficiency, requiring only $1/15$ of the paired data for fine-tuning compared to a baseline trained on the full dataset.

(iii) *Scalable high-resolution generation.* GMem maintains strong performance at $1024 \times 1024$ resolution, demonstrating that the efficiency advantages extend beyond moderate resolutions to challenging high-resolution generation tasks.

These results establish GMem as a data- and compute-efficient approach for T2I generation and unified multimodal models, proving its potential across diverse network architectures and resolutions.

### 5.5 ABLATION STUDIES

The performance of GMem depends on several factors: bank size, backbone architecture, solver, and masking strategies. We conduct ablation studies on ImageNet $256 \times 256$ with 64 epochs, and found that while each factor slightly affects optimal performance, GMem consistently generates high-quality images efficiently.

- **Memory capacity scaling.** As shown in Table 2 , we sweep memory capacity $N$ from 1.2M to 1000 entries during sampling, observing monotonic FID improvements with larger banks but diminishing returns at scale; detailed analysis is provided in Appendix G.5 .

- **Memory-parameter trade-offs.** Experiments also reveal that 2K memory snippets can substitute for approximately 1M trainable parameters while maintaining comparable generation quality; see Appendix G.5 for quantitative analysis.

- **Additional findings.** We also find that moderate masking ratios (0.4) achieve optimal performance, SVD-based compression reduces memory size while improving FID, SDE solvers consistently outperform ODE solvers, and GMem generalizes well across various backbones and tokenizers; comprehensive results are provided in Appendix G.5 .

## 6 CONCLUSION

In this work, we systematically introduce an explicit memorization mechanism into the Diffusion Transformer, proposing a novel framework named GMem. This design yields a $50 \times$ acceleration in training speed on the ImageNet $256 \times 256$ dataset. Furthermore, we demonstrate that a GMem model pretrained on ImageNet can serve as a powerful general-purpose base model. It can be efficiently adapted to diverse downstream tasks, such as anime, face, and medical imaging, with only $\sim 20$K fine-tuning steps, achieving performance comparable to training from scratch. Finally, we present a data- and compute-efficient pipeline for text-to-image generation. This pipeline achieves performance on par with the PixArt-$\alpha$ baseline (GenEval score: 0.52 vs. 0.48) while using only $1/17$ of the data and $1/9$ of the total training time.

## ETHICS STATEMENT

This work complies with the ICLR Code of Ethics. Our research did not involve human subjects or animal experimentation. All datasets utilized in this study, including ImageNet (Deng et al., 2009), CIFAR-10 (Krizhevsky et al., 2009), the OpenUni Dataset (Wu et al., 2025), and Micro-diffusion (Sehwag et al., 2024), are publicly available and have been appropriately cited. We have taken rigorous measures to mitigate potential biases and prevent discriminatory outcomes. Furthermore, our research did not involve any personally identifiable information, and our experiments were designed to pose no privacy or security risks. We are dedicated to upholding the principles of transparency and integrity throughout our research process.

## REPRODUCIBILITY STATEMENT

We are committed to ensuring the reproducibility of the results presented in this paper. To this end, all source code and datasets are publicly available in the supplementary materials. The paper provides a detailed description of the experimental setup, including training procedures, model configurations, and hardware specifications. To further facilitate the replication of our experiments, we have also provided comprehensive implementation details in Appendix F . We are confident that these measures will enable other researchers to verify our findings and build upon our work, thereby contributing to the advancement of the field.

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

CONTENTS

## A   USE OF LLMS

This paper only uses LLMs for polishing.

## B   RELATED WORK

**External representation-augmented diffusion models via retrieval and generation.**   As the sampling process of GMem follows a Retrieval-Augmented Generation (RAG) manner, we briefly review the RAG methods in generative models. Retrieval-Augmented Generation (RAG) enhances generation quality by integrating external knowledge. IC-GAN (Casanova et al., 2021) augments image generation by conditioning on neighborhood instances retrieved from the training dataset. However, using only training images limits generalization. To address this issue, KNN-Diffusion (Sheynin et al., 2022) and RDM (Blattmann et al., 2022) employ large external memory sets, guiding generation via kNN retrieval during training and inference. Similarly, Chen et al. (2022) and Li et al. (2022) leverage a set of text-image pairs with cross-modality retrieval, improving generation performance on rare images.

Despite their advantages, RAG methods encounter two key challenges: (i) substantial storage demands for large memory sets, and (ii) increased computational costs during retrieval. We further contend that over-reliance on training sets restricts generalization capabilities (Blattmann et al., 2022). By employing a masking strategy (see our Appendix G.5), we mitigate this dependency without incurring additional computational or storage overhead, thereby improving both generalization and training/inference efficiency.

Representations for generation augmentation can also be obtained from representation generators. For instance, RCG (Li et al., 2023b) employs a representation generator to produce 1D "memory snippets" to guide diffusion models. Although RCG reduces the need to store large-scale memory sets, it encounters two primary challenges: (i) the requirement for additional training and sampling processes for the representation generator, increasing computational demands; (ii) the necessity to retrain the representation generator to incorporate knowledge of new classes or artistic style transfers, thereby limiting its generalization capability. To overcome these limitations, Appendix D.1 presents an efficient, training-free method for incorporating additional knowledge into the memory bank.

## C   STORAGE-EFFICIENT DECOMPOSITION STRATEGY

However, akin to Blattmann et al. (2022); Casanova et al. (2021), the memory bank requires explicitly storage and retrieval during inference. For large-scale datasets like ImageNet, this results in prohibitive storage demands and retrieval costs. Additionally, memory snippets often exhibit significant redundancy (e.g., snippets from the same class are highly correlated), leading to unnecessary storage when all features are retained directly.

To reduce redundancy and minimize storage, we propose an efficient storage strategy based on matrix decomposition. The core idea is to represent the large memory bank $\mathbf{M} \in \mathbb{R}^{N \times d}$ by three much smaller components: a mean vector $\boldsymbol{\mu}$, a coefficient matrix $\mathbf{C}$, and a basis matrix $\mathbf{B}$. This decomposition is achieved through the following three-step process:

 (i) **Center the Memory Bank.** First, we compute the mean snippet $\boldsymbol{\mu} = \mathrm{mean}(\mathbf{M}) \in \mathbb{R}^{1 \times d}$ from the full memory bank $\mathbf{M}$. We then center the bank by subtracting this mean from every snippet, yielding a centered matrix $\mathbf{M}_c = \mathbf{M} - \boldsymbol{\mu}$.

 (ii) **Apply Truncated SVD.** Next, we apply Singular Value Decomposition (SVD) to the centered matrix $\mathbf{M}_c$ and truncate it to a rank $r$ (e.g., $r = 512$). This factorizes the matrix as $\mathbf{M}_c \approx \mathbf{U}_r \boldsymbol{\Sigma}_r \mathbf{V}_r^\top$, where $\mathbf{U}_r, \boldsymbol{\Sigma}_r, \mathbf{V}_r$ contain the top $r$ components.

(iii) **Form the Compact Representation.** Finally, we use the SVD factors to define our compact storage components. We construct the coefficient matrix $\mathbf{C} = \mathbf{U}_r \boldsymbol{\Sigma}_r^{1/2} \in \mathbb{R}^{N \times r}$ and the basis matrix $\mathbf{B} = \mathbf{V}_r \boldsymbol{\Sigma}_r^{1/2} \in \mathbb{R}^{d \times r}$.

The matrix $\mathbf{B}$ acts as a compact, fixed basis encoding global structure, whereas $\mathbf{C}$ flexibly stores snippet-specific coefficients. By storing $\mathbf{C}$ and $\mathbf{B}$ separately we achieve compressed storage, reducing the space cost from $\mathcal{O}(Nd) \to \mathcal{O}(Nr + dr)$.

During inference, retrieving a snippet involves looking up its coefficients in $\mathbf{C}$ and transforming them via $\mathbf{B}$. Specifically, the $i$-th memory snippet $\mathbf{s}_i$ can be reconstructed as:

$$\mathbf{s}_i = \mathbf{c}_i \mathbf{B}^\top + \boldsymbol{\mu} \,, \tag{2}$$

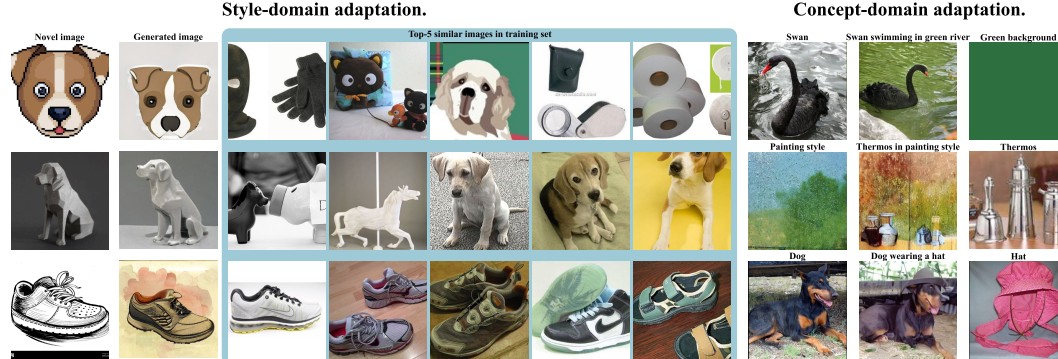

Figure 5: **Demonstration of test-time style and concept-domain adaptation via memory manipulation.** Selected samples from ImageNet $256 \times 256$ generated by the GMem. In the "Style-domain adaptation." part, we show the reference image used to build a new snippet (left), followed by the generated samples and 5 of the nearest training images, illustrating GMem's adaptation to external memory. In the "Concept-domain adaptation." examples, two reference images (left and right) form an interpolated image (center), demonstrating GMem can manipulate internal memory to create new concepts.

where $\mathbf{c}_i \in \mathbb{R}^{1 \times r}$ is the coefficient vector corresponding to the $i$-th snippet from coefficient matrix $\mathbf{C}$.

## D  MEMORY MANIPULATION ENABLES TEST-TIME ADAPTATION

### D.1  METHODS

The core insight of GMem lies in introducing an explicit memory bank to generative modeling, which allows us to manipulate the memory bank to enabling generation beyond training data. This is achieved through two approaches: (i) incorporating external memory by introducing novel images absent from the original dataset, and (ii) manipulating internal memory by combining existing snippets into new compositions. While our memory bank provides compact storage and flexibility for new snippets integration, we note that advancing the modularity of the memory bank remains future work.

**Aspect I: external memory augmentation.** To incorporate external memory outside the training dataset, we project the feature vector $\boldsymbol{f}(\mathbf{x}_{\text{new}}) \in \mathbb{R}^d$ onto the existing coefficient matrix $\mathbf{C}$. We calculate the coefficients $\mathbf{c}_{\text{new}} \in \mathbb{R}^r$ of the centered feature vector by projecting it onto the basis matrix $\mathbf{B}$:

$$\mathbf{c}_{\text{new}} = (\boldsymbol{f}(\mathbf{x}_{\text{new}}) - \boldsymbol{\mu})\, \mathbf{B}/\mathbf{S}\,, \tag{3}$$

where $\mathbf{S}$ (the diagonal of $\boldsymbol{\Sigma}$) stores singular values[1]. By appending $\mathbf{c}_{\text{new}}$ to $\mathbf{C}$, we expanding the memory bank with negligible overhead. This process seamlessly integrates new snippets not present in the training dataset, allowing the model to utilize the network's generalization capabilities for generating new samples without additional training.

**Aspect II: internal memory modification.** We generate new memory snippets by interpolating between existing ones. Given two snippets indexed by $i$ and $j$, we construct a new coefficient vector:

$$\mathbf{c}_{\text{new}} = \alpha\, \mathbf{c}_i + (1 - \alpha)\, \mathbf{c}_j, \quad \alpha \in [0, 1]\,. \tag{4}$$

Appending $\mathbf{c}_{\text{new}}$ to $\mathbf{C}$ yields a latent interpolation in $\mathbf{M}$ without modifying $\mathbf{B}$. This approach enables training-free style transfer and compositional generalization by exploring linear paths between coefficients of different memory snippets, effectively creating novel samples from the internal memory encoded in $\mathbf{C}$.

### D.2  RESULTS

**Concept-composition and image editing.**  Beyond distributional shift, we examine compositionality and editability at inference without any additional training. We show that GMem can adjust to domains never encountered during training simply by manipulating its external memory at inference time. Two complementary cases are considered below:

---

[1]These singular values are several floating-point numbers with negligible storage cost.

(i) ***style-domain adaptation via memory augmentation:*** As illustrated in Section D.1 , a single snippet extracted from an unseen reference image (for example a *low-poly* or *charcoal sketch* photograph) is appended to the memory bank. Without updating network weights, GMem can render similar images in the new style while preserving their semantics. To be specific, as shown in Figure 5 , the generated samples inherit the characteristic contours and shading of the reference style yet remain visually distinct from it, indicating genuine synthesis rather than direct copying.

(ii) ***concept-domain adaptation via memory modification:*** As illustrated in Section D.1 , two existing snippets that encode known concepts such as *dog* and *hat* are combined into a novel-concept snippet. With the modified snippet, GMem produces coherent hybrids like a *dog wearing a hat*, illustrated in Figure 5 . This approach also allows introducing artistic concepts to existing classes, e.g., we can generate a *swan swimming in a green river* by interpolating between a internal snippet *swan* and a external concept *green background* snippet. The resulting images suggesting that the network successfully adapt to new concept-domain on the fly.

These qualitative results further confirm that GMem can achieve test-time domain adaptation (both stylistic transfer and concept composition) without any retraining. We argue that such controllable interpolation highlights GMem 's potential for real-time, user-guided image editing workflows.

## E   BRIDGING GMEM AND DIFFUSION TRANSFORMERS

### E.1   REDUCTION GMEM TO DIFFUSION TRANSFORMERS.

The external memory in GMem can be regarded as a class–conditioned embedding table: each label $y$ is linked to a set of snippet vectors. Appendix F.8 shows that keeping only one–tenth of the original bank (about 130 snippets per ImageNet class) incurs a minor FID increase. This observation suggests a training-free procedure for gradually reducing GMem into a standard diffusion transformer:

(i) **Group** all snippets by their image class.

(ii) **Reduce** each class to $k \in \{13, 5, 1\}$ representatives, denoted Random@k (random pick) or Average k (snippet average).

(iii) **Initialise** the network's `label_embedder` with the resulting $k$ embeddings per class and sample without further optimisation.

Table 4 summarises the outcome. With Random@1 using LightningDiT-B/1+REPA+GMem— exactly one embedding per class, matching LightningDiT setting, FID rises from 6.96 to 8.92 yet remains comparable the LightningDiT-B/1 baseline (15.82). Intermediate settings (Random@5, Random@13) offer a smooth quality–memory trade-off.

**Which reduction to prefer?**   For a well-trained GMem (FID<6) we recommend Random@1: selecting a real snippet preserves fine semantics, whereas averaging may blur details and harm quality. For earlier checkpoints, a coarse yet representitive Average@1 can suppress noisy snippets and often yields slightly better FID.

Table 4: **Model Comparison on Random Sampling Metrics**.  All models are evaluated under standard configurations. ↓ indicates lower is better.

| Model | Average@1 ($\downarrow$) | Random@1 ($\downarrow$) | Random@5 ($\downarrow$) | Random@13 ($\downarrow$) |
|---|---|---|---|---|
| LDiT-B/1 + REPA + GMem | 8.92 | 10.62 | 7.75 | 7.48 |
| LDiT-XL/1 + REPA + GMem | 12.27 | 7.62 | 4.15 | 3.41 |
| SiT-XL/2 + REPA + GMem | 6.54 | 12.90 | 5.96 | 5.41 |

### E.2   PROJECTION FROM DIFFUSION TRANSFORMERS TO GMEM

Starting from a converged SiT-XL+REPA checkpoint ($4\,$M pre-training steps), we fine-tune for 20K steps while enabling the GMem framework described in Section 4 . Figure 8 shows the model adapts rapidly: after 10K steps—roughly $0.25\%$ per cent of pretrainig step—it recovers comparable FID=5.52 against pretraining checkpoint. By 20K fine-tuning steps, the model reaching the same FID obtained when GMem is trained for 400K steps.

---

**Algorithm 1** Training GMem using memory bank $\mathbf{M}$

---

**procedure** TRAIN GMEM($\mathbf{v}_\theta, \mathcal{D}, \mathbf{M}, T, \alpha_t, \sigma_t$)
    **Initialize** model parameters $\theta$
    **for** each training iteration **do**
        **Sample** a batch of data $\mathbf{x}_0 \sim \mathcal{D}$
        **Sample** timesteps $t \sim \{0, \ldots, T\}$ uniformly
        **Generate** noise $\boldsymbol{\epsilon} \sim \mathcal{N}(0, \mathbf{I})$
        **Compute** noisy data $\mathbf{x}_t = \alpha_t \mathbf{x}_0 + \sigma_t \boldsymbol{\epsilon}$
        **Sample** memory snippets $\mathbf{s} \sim \mathbf{M}$
        **Mask** $\mathbf{s} \leftarrow \mathbf{s} \odot$ mask (in feature dimension)
        **Predict** velocity $\mathbf{v}_\theta(\mathbf{x}_t, t, \mathbf{s})$
        **Compute** loss using (1)
        **Backpropagate** and update $\theta$ using Optimizer
    **end for**
**end procedure**

---

A common concern is whether the fine-tuned model still supports test-time domain adaptation. We evaluate this using the style-transfer protocol of Appendix G.7 , where lower LPIPS indicates better adherence to the reference style. Table 13 shows that after 20K steps the retrofitted model (REPA$_{4M}$+GMem) achieves LPIPS scores superior to the ImageNet and virtually identical to the fully trained GMem. Thus, memory fine-tuning restores both image quality and adaptation ability with negligible computational overhead.

# F  IMPLEMENTATION DETAILS

## F.1  COMPUTING RESOURCES

All models are primarily trained on NVIDIA H800 8-GPU setups, each equipped with 80GB memory and WCT is measured based on such setup.

## F.2  EXTRA METRICS

We categorize the metrics into three groups. *(i) quality:* we follow (Dhariwal & Nichol, 2021) and report FID-50K (FID) (Heusel et al., 2017). *(ii) diversity:* diversity is assessed with LPIPS (Zhang et al., 2018), SSIM (Wang et al., 2004), and Peak signal-to-noise ratio (PSNR), computed between each generated image and its nearest neighbours in the training set. We argue that under good generating quality (guaranteed by lower FID), methods with lower SSIM, PSNR and higher LPIPS have better diversity. *(iii) efficiency:* training cost is measured in epochs[2], while sampling cost is measured in the number of function evaluations (NFE). We also quote wall-clock time (WCT), measured in minutes, to indicating the time taken to sample 50K images for completeness.

## F.3  EXTRA BASELINES

For a fair comparison, we compare to the SoTA image generation methods on both training efficiency and performance. Specifically, for pixel-space image generation, we consider the following three categories of baselines: First, we compare GMem with traditional generative models, including Diffusion GAN (Xiao et al., 2021), Diffusion StyleGAN (Wang et al., 2022a), DMD2 (Yin et al., 2024). We also compare the SoTA diffusion models with UNets, including DDPM (Ho et al., 2020), Score SDE (Song et al., 2020b), EDM (Karras et al., 2022), DPM-Solver (Lu et al., 2022), ADM (Dhariwal & Nichol, 2021), EDMv2 (Karras et al., 2024), CTM (Kim et al., 2023), SiD (Zhou et al., 2024). Finally, we also compare to the SoTA flow-based transformer methods, including DiT (Peebles & Xie, 2023), SiT (Ma et al., 2024), and the most recent yet contemporaneous work REPA (Yu et al., 2024).

---

[2]64 epochs correspond to $\sim$ 80K steps at batch size 1024; results obtained with other settings are rescaled accordingly.

### F.4 LATENT DIFFUSION MODEL TRAINING FRAMEWORK

We closely follow the training protocol used in REPA (Yu et al., 2024) and SiT (Ma et al., 2024). Similar to a Vision Transformer (Dosovitskiy et al., 2021), In this architecture, the input image is divided into patches, reshaped into a one-dimensional sequence of length $N$, and then processed by the model.

For latent space generation, SiT uses a downsampled latent image $z = E(x)$ as input, where $x$ is an RGB image and $E$ is the vision tokenizer of the Stable Diffusion Variational Autoencoder (VAE) (?). For pixel space generation, we remove the vision tokenizer and directly use the RGB image as input. Specifically, we modify the original SiT by changing the number of channels from 4 to 3 and directly feed the transformed RGB image into the model.

### F.5 HYPERPARAMETERS

We detail the hyperparameter configurations for experiments based on different backbones in Table 5 and Table 6.

**Experiments with SiT backbone.**  For experiments using SiT (Ma et al., 2024) as the backbone, we follow the hyperparameter settings of the original REPA implementation (Yu et al., 2024) to ensure consistency and fair comparison. Specifically, we adopt the AdamW optimizer (Kingma & Ba, 2014) with a constant learning rate of $1 \times 10^{-4}$, $\beta_1 = 0.9$, and $\beta_2 = 0.999$, and no weight decay. To accelerate training, we use mixed-precision (fp16) computation and apply gradient clipping. For latent space generation, we pre-compute compressed latent vectors from raw images using SD-VAE (?), which are then used as input. For pixel space generation, we directly feed raw pixel data.

**Experiments with LightningDiT backbone.**  For experiments using LightningDiT (Yao & Wang, 2025) as the backbone, we implement GMem based on the official LightningDiT codebase and strictly follow its hyperparameter setup. The main differences between LightningDiT-based setup and the SiT-based setup lie in the batch size, learning rate, and patch size. GMem uses a larger batch size of 1024 and a higher learning rate of $2 \times 10^{-4}$ with AdamW optimizer ($\beta_1 = 0.9$, $\beta_2 = 0.95$), consistent with LightningDiT. Moreover, to be compatible with VA-VAE-f32d32 (Yao & Wang, 2025), we use a patch size of 1, ensuring a fixed sequence length of $N = 256$.

### F.6 MEMORY PROJECTION

For projecting memory snippets into the backbone hidden dimension, we utilize a three-layer MLP with SiLU activations for diffusion transformers, following Yu et al. (2024).

### F.7 VISION ENCODER

We adopt a unified vision encoder, Dinov2-B (Oquab et al., 2023), across all our experiments to facilitate more effective representation learning. This choice brings two key advantages. First, it has been shown to significantly enhance the learning of better representations in diffusion models (Yu et al., 2024). Second, using a consistent encoder across tasks enables the memory bank to perform zero-shot knowledge transfer across different datasets more effectively, as described in Appendix H.4.

### F.8 MEMORY BANK.

We use a memory bank of size 50K for CIFAR-10 and 1.28M for ImageNet $256 \times 256$ and ImageNet $512 \times 512$.

To reduce the memory overhead, we further explore two strategies that enable significant memory bank size reduction with minimal impact on performance. First, as shown in Figure 4 and Table 11, we find that reducing the memory bank size by $10\times$ only results in a minor FID increase of $0.25\%$, and a $2\times$ reduction leads to an almost negligible average increase of $0.08\%$ in FID. Second, we propose an SVD-based compression method to further lower the memory bank cost without substantial degradation in generation quality.

## G ADDITIONAL EXPERIMENTS

In this section, we include a supplementary experiments that apply GMem to further validate the effectiveness of GMem on various downstream tasks.

Table 5: **Training settings of CIFAR-10.** We provide the training settings for all models and training algorithms on the CIFAR-10 dataset.

|  | Model Size | | |
|---|---|---|---|
|  | B | L | XL |
| **Model details** | | | |
| Batch size | 128 | 128 | 128 |
| Training iterations | 200K | 200k | 200k |
| Learning rate | 1e-4 | 1e-4 | 1e-4 |
| Optimizer | Adam | Adam | Adam |
| Adam $\beta_1$ | 0.9 | 0.9 | 0.9 |
| Adam $\beta_2$ | 0.999 | 0.999 | 0.999 |
| **Interpolants** | | | |
| $\alpha_t$ | $1-t$ | $1-t$ | $1-t$ |
| $\sigma_t$ | $t$ | $t$ | $t$ |
| $\omega_t$ | $\sigma_t$ | $\sigma_t$ | $\sigma_t$ |
| Training Objective | v-prediction | v-prediction | v-prediction |
| Sampler | Euler | Euler | Euler |
| Sampling steps | 50 | 50 | 50 |
| Classifier-free Guidance | $\times$ | $\times$ | $\times$ |
| **Training details of backbone** | | | |
| Capacity(Mparams) | 130 | 458 | 675 |
| Input dim. | $32\times32\times3$ | $32\times32\times3$ | $32\times32\times3$ |
| Num. layers | 12 | 24 | 28 |
| Hidden dim. | 768 | 1,024 | 1,152 |
| Num. heads | 12 | 12 | 16 |
| **Training details of GMem** | | | |
| Bank size | 50k | 50k | 50k |
| Encoder $f(\mathbf{x})$ | DINOv2-B | DINOv2-B | DINOv2-B |

Table 6: **Training settings for LightningDiT-based GMem.** We present the training settings for all models and training algorithms on the ImageNet $256 \times 256$ dataset (left) and the ImageNet $512 \times 512$ dataset (right).

|  | Model Size | | |  |  | Model Size | | |
|---|---|---|---|---|---|---|---|---|
|  | B | L | XL |  |  | B | L | XL |
| **Model details** | | | | | **Model details** | | | |
| Batch size | 1024 | 1024 | 1024 | | Batch size | 1024 | 1024 | 1024 |
| Training iterations | 200K | 80K | 600M00K | | Training iterations | 500K | 500k | 500k |
| Learning rate | 2e-4 | 2e-4 | 2e-4 | | Learning rate | 1e-4 | 1e-4 | 1e-4 |
| Optimizer | Adam | Adam | Adam | | Optimizer | Adam | Adam | Adam |
| Adam $\beta_1$ | 0.9 | 0.9 | 0.9 | | Adam $\beta_1$ | 0.9 | 0.9 | 0.9 |
| Adam $\beta_2$ | 0.995 | 0.995 | 0.995 | | Adam $\beta_2$ | 0.995 | 0.995 | 0.995 |
| Image tokenizer | VA-VAE | VA-VAE | VA-VAE | | Image tokenizer | DC-AE | DC-AE | DC-AE |
| **Interpolants** | | | | | **Interpolants** | | | |
| $\alpha_t$ | $1-t$ | $1-t$ | $1-t$ | | $\alpha_t$ | $1-t$ | $1-t$ | $1-t$ |
| $\sigma_t$ | $t$ | $t$ | $t$ | | $\sigma_t$ | $t$ | $t$ | $t$ |
| $\omega_t$ | $\sigma_t$ | $\sigma_t$ | $\sigma_t$ | | $\omega_t$ | $\sigma_t$ | $\sigma_t$ | $\sigma_t$ |
| Training objective | v-prediction | v-prediction | v-prediction | | Training objective | v-prediction | v-prediction | v-prediction |
| Sampler | Heun | Heun | Heun | | Sampler | Heun | Heun | Heun |
| Sampling steps | 100 | 100 | 100 | | Sampling steps | 100 | 100 | 100 |
| **Training details of backbone** | | | | | **Training details of backbone** | | | |
| Capacity (Mparams) | 130 | 458 | 675 | | Capacity(Mparams) | 130 | 458 | 675 |
| Num. layers | 12 | 24 | 28 | | Num. layers | 12 | 24 | 28 |
| Hidden dim. | 768 | 1,024 | 1,152 | | Hidden dim. | 768 | 1,024 | 1,152 |
| Num. heads | 12 | 12 | 16 | | Num. heads | 12 | 12 | 16 |
| **Training details of GMem** | | | | | **Training details of GMem** | | | |
| Bank size | 1.2M | 1.2M | 1.2M | | Bank size | 1.28M | 1.28M | 1.28M |
| Encoder $f(\mathbf{x})$ | DINOv2-B | DINOv2-B | DINOv2-B | | Encoder $f(\mathbf{x})$ | DINOv2-B | DINOv2-B | DINOv2-B |

## G.1 PRIVACY RISK WHEN USING EXTERNAL MEMORY

**Experimental setup.** We consider a conservative threat model in which both a single memory snippet **s** and the released GMem checkpoint (ImageNet $256 \times 256$, 650 epochs) are exposed to an adversary. No further training is performed and the memory bank is disabled during evaluation. To assess reconstruction risk, we adopt three widely used image inversion metrics: SSIM ($\uparrow$, structural similarity), PSNR ($\uparrow$, dB, pixel fidelity), and LPIPS ($\downarrow$, perceptual distance). Following prior work, we regard reconstructions as visually similar if they satisfy SSIM $\geq 0.50$, PSNR $\geq 20$ dB, and LPIPS $\leq 0.30$. The testing procedure is as follows: (i) generate images conditioned directly on the exposed snippet using the bank-free generator, (ii) retrieve the nearest neighbor from the training set via 1-NN

Table 7: **Training settings of SiT-based GMem.** We provide the training settings for all models and training algorithms on the ImageNet $256 \times 256$ dataset.

| | Model Size | | |
| --- | --- | --- | --- |
| | **B** | **L** | **XL** |
| **Model details** | | | |
| Batch size | 256 | 256 | 256 |
| Training iterations | 200K | 400k | 400k |
| Learning rate | 1e-4 | 1e-4 | 1e-4 |
| Optimizer | Adam | Adam | Adam |
| Adam $\beta_1$ | 0.9 | 0.9 | 0.9 |
| Adam $\beta_2$ | 0.999 | 0.999 | 0.999 |
| Image tokenizer | SD-VAE | SD-VAE | SD-VAE |
| **Interpolants** | | | |
| $\alpha_t$ | $1 - t$ | $1 - t$ | $1 - t$ |
| $\sigma_t$ | $t$ | $t$ | $t$ |
| $\omega_t$ | $\sigma_t$ | $\sigma_t$ | $\sigma_t$ |
| Training Objective | v-prediction | v-prediction | v-prediction |
| Sampler | Euler | Euler | Euler |
| Sampling steps | 250 | 250 | 250 |
| Classifier-free Guidance | $\times$ | $\times$ | $\times$ |
| **Training details of backbone** | | | |
| Capacity(Mparams) | 130 | 458 | 675 |
| Num. layers | 12 | 24 | 28 |
| Hidden dim. | 768 | 1,024 | 1,152 |
| Num. heads | 12 | 12 | 16 |
| **Training details of GMem** | | | |
| Bank size | 1.28M | 1.28M | 1.28M |
| Encoder $f(\mathbf{x})$ | DINOv2-B | DINOv2-B | DINOv2-B |

search, and (iii) compute SSIM, PSNR, and LPIPS between the generated image $\hat{x}$ and its nearest neighbor $x^*$.

Table 8: **Privacy risk evaluation under snippet exposure.** We report reconstruction quality when exposing a single DINOv2 [CLS] memory snippet and the pretrained GMem checkpoint. Metrics include SSIM ($\uparrow$), PSNR ($\uparrow$, dB), and LPIPS ($\downarrow$). Thresholds for visually similar reconstructions are SSIM $\geq 0.50$, PSNR $\geq 20$, and LPIPS $\leq 0.30$. GMem shows no reconstruction capability and exhibits risk comparable to a standard diffusion baseline.

| Model | SSIM ($\uparrow$) | PSNR ($\uparrow$) | LPIPS ($\downarrow$) |
| --- | --- | --- | --- |
| VA-VAE-f16d32 | 0.79 | 27.96 | 0.10 |
| SiT-XL/2 + REPA (1400 epochs) | 0.16 | 9.46 | 0.70 |
| **GMem (650 epochs)** | **0.17** | **9.58** | **0.70** |

**Results.** Table 8 reports reconstruction similarity. GMem achieves SSIM $= 0.17$, PSNR $= 9.58$, and LPIPS $= 0.70$—all far below the operational thresholds for visual similarity, comparable to a standard diffusion baseline (SiT-XL/2 + REPA, 0.16 / 9.46 / 0.70) and significantly different from a VAE-based model (VA-VAE-f16d32, 0.79 / 27.96 / 0.10). These findings indicate that under the evaluated threat model—exposure of a DINOv2 [CLS] memory snippet and a released checkpoint—GMem does not enable reconstruction of training images and does not introduce privacy risks beyond those of conventional diffusion models.

**Related works.** Our empirical findings also align with recent theoretical and empirical analyses of DINOv2 representations. Specifically, the [CLS] token, while semantically informative, has been shown to lack sufficient low-level spatial detail for accurate image reconstruction. First, Jose et al. (2025) report that using the [CLS] token alone for zero-shot segmentation yields very low mIoU ($\sim 8.3\%$), whereas appending patch-average features substantially increases performance ($\sim 18.2\%$). This gap suggests that [CLS] omits fine-grained details critical for dense prediction tasks. Second, Darcet et al. (2023) demonstrate that high-norm "global" tokens—functionally analogous to [CLS]—are significantly less informative about local pixel values and spatial positions than patch

Table 9: **Efficiency and diversity of GMem**. *(i)* $50\times$ *training speedup:* GMem delivers a $50\times$ reduction in training epochs versus the SiT baseline on ImageNet $256 \times 256$. *(ii)* $10\times$ *sampling speedup:* with the same SiT-L/2 backbone, GMem reaches the target FID in as few as NFE=25, cutting sampling cost by $10\times$. *(iii) Diversity:* GMem, based on LightningDiT-XL/1 with REPA loss, retains the diversity of generated images while generating better quality images.

| Method | Epoch (↓) | FID (↓) |
|---|---|---|
| SiT-XL/2 | 1400 | 8.61 |
| + REPA | 800 | 5.90 |
| + GMem | 80 | 5.27 |
| LightningDiT-XL/1 | 800 | 2.17 |
| + REPA | 160 | 1.84 |
| + GMem | 160 | 1.53 |

| Model | Epoch | NFE (↓) | WCT (↓) | FID (↓) |
|---|---|---|---|---|
| SiT-XL/2 + REPA | 20 | 250 | 70 | 5.9 |
| SiT-L/2 | 20 | 250 | - | 18.8 |
| + REPA | 20 | 250 | 40 | 8.4 |
| + GMem | 20 | 250 | 42 | **5.8** |
| + GMem | 20 | 50 | 12 | **7.5** |
| + GMem | 20 | 25 | 6 | **12.3** |

| Model | Epoch | FID | SSIM (↓) | PSNR (↓) | LPIPS |
|---|---|---|---|---|---|
| LDM-f16d16 | - | 0.49 | 0.72 | 26.10 | 0.13 |
| DC-AE-f64d128 | - | 0.81 | 0.65 | 23.60 | 0.09 |
| VA-VAE-f16d32 | - | 0.28 | 0.79 | 27.96 | 0.10 |
| SiT-XL/2 + REPA | 64 | 8.42 | 0.18 | 9.39 | 0.71 |
| SiT-XL/2 + REPA | 1400 | 5.94 | 0.16 | 9.46 | 0.70 |
| GMem | 650 | 1.43 | 0.17 | 9.58 | 0.70 |

tokens, further undermining their utility for image reconstruction. Together, these results support our conclusion that exposing memory snippets based on `[CLS]` tokens does not materially elevate privacy risk beyond a standard diffusion baseline.

### G.2 DIVERSITY OF GENERATED SAMPLES

**GMem does not hurt sample diversity when integrating it into diffusion transformers.** To verify that GMem does *not* induce mode collapse, we compare the diversity of images generated by GMem and a vanilla DiT baseline (i.e. REPA). Despite reaching a SoTA FID of $1.43$, GMem matches the baseline on all three perceptual metrics, indicating unchanged diversity.[3] In addition to the quantitative metrics, we also provide a qualitative comparison in 10, where we visualize per-class grids from the training set, the REPA baseline, and GMem. We observe that both GMem and REPA preserve the target class semantics while exhibiting diverse backgrounds, poses, and fine-grained appearance variations comparable to those in the real data, further confirming that GMem does not hurt sample diversity.

### G.3 SAMPLING EFFICIENCY

Beyond training efficiency, GMem also improves inference. Specifically, we show that it accelerates sampling by (i) enabling smaller networks to match the quality of larger ones, and (ii) reducing the NFEs required to reach a given FID target.

(i) *a smaller network for comparable FID:* fixing SiT-L+REPA as the network and matching the FID of SiT-XL+REPA, GMem executing each step on a network that has roughly half the parameters (458M and 675M). The resulting wall-clock time is reduced by $1.66\times$, which reflects the much lighter network used in GMem. Details are listed in Table 9.

(ii) *fewer NFEs using the same network:* using the SiT-L+REPA network, we compare GMem against two baselines: the original SiT-L and SiT-L+REPA. As summarised in Table 9, GMem reaches the target FID=8.4 with only NFE=50: a $5\times$ reduction relative to REPA and still keeps a lower FID=12.3 when NFE is further reduced to 25 (a $10\times$ speed-up compared to SiT-L). Wall-clock measurements also suggest a $\mathbf{10\times}$ **speed-up** than SiT-L.

It is worth mentioning that in the sampling experiments, we used the same sampler (DDIM) and the same hyperparameters for all methods, as detailed in Table 7. These two observations show that GMem can deliver inference acceleration by allowing a smaller network to reach the same image quality or by shortening the diffusion trajectory.

### G.4 HIGH-RESOLUTION IMAGE GENERATION

We conduct additional experiments on the image generation benchmark using the FFHQ $1024 \times 1024$ dataset.

**Model Architecture.** We employ *LightningDiT-XL/1 + DC-AE-f32d32* as the backbone architecture, starting from a model pretrained on ImageNet at $512 \times 512$ resolution. To accommodate the resolution increase, we double the patch size of LightningDiT-XL from 1 to 2, maintaining the same number of input tokens as the original $512 \times 512$ training setup.

---

[3]Full quantitative results are listed in Table 9; qualitative samples are visualised in Figure 9.

**Training Configuration.**    We fine-tune the pretrained model (trained for 200 epochs on ImageNet $512 \times 512$) on the FFHQ $1024 \times 1024$ training split for 20K steps. All training hyperparameters—including batch size, learning rate, optimizer, and data augmentation strategies—remain consistent with the pretraining stage. The entire pretrained model undergoes full fine-tuning.

**Evaluation Protocol.**    We evaluate generation quality using FID with 10K generated samples following (Zhang et al., 2023). We compare our approach against DVDP trained from scratch on FFHQ $1024 \times 1024$ as a representative baseline for high-resolution image generation.

Table 10: **High-resolution image generation on FFHQ** $1024 \times 1024$. We compare GMem with DVDP and Score-SDE (Song et al., 2020b). **Training budget** is measured in epochs, and **NFE** denotes the number of function evaluations during sampling.

| Method | Training budget (Epoch) | NFE | FID |
|---|---|---|---|
| GMem | 0.3 (finetuning) | $50 \times 2$ | 11.57 |
| DVDP (Zhang et al., 2023) | — | 1000 | 10.46 |
| Score-SDE (Song et al., 2020b) | 274 | 2000 | 52.40 |

**Results.**    We list the main results in Table 10. **Cross-dataset generalization and resolution transfer.** The successful adaptation from ImageNet $512 \times 512$ pretraining to FFHQ $1024 \times 1024$ generation with only 20K fine-tuning steps showcases GMem's cross-dataset generalization capabilities. **High-resolution generation capability.** GMem achieves an FID of 11.57 on FFHQ $1024 \times 1024$. When compared to DVDP's competitive performance of 10.46 FID, GMem demonstrates comparable high-resolution generation quality while using much less training and sampling budget (0.3 epochs vs. 274 epochs, $50 \times 2$ NFE vs. 2000 NFE).

This efficient transfer—requiring minimal additional training—demonstrates that GMem can rapidly adapt from low-resolution to high-resolution generation across different domains (natural images to faces). We hope this could inspire new ideas on leveraging pretrained models to enhance the efficiency of high-resolution image generation.

### G.5    MORE ABLATION STUDIES

Table 11: **Ablation study and sensitivity analysis**. All models are trained on ImageNet $256 \times 256$ without classifier-free guidance. Unless otherwise specified, the backbone is LightningDiT-B/1 (LDiT-B/1), the vision encoder is DINOv2-B, and training runs for 64 epochs. ↓ indicates lower is better.

| Epochs | Backbone | Vision Encoder | Bank size | SVD | Mask Strategy | Solver | FID (↓) |
|---|---|---|---|---|---|---|---|
| 64 | LDiT-B/1 | DINOv2-B | 1.2M | ✓ | (Zero, 0.4) | SDE | **5.70** |
| 64 | LDiT-B/1 | DINOv2-B | 1.2M | × | (Zero, 0.4) | SDE | 5.85 |
| 64 | LDiT-B/1 | DINOv2-B | 640K | ✓ | (Zero, 0.4) | SDE | 5.72 |
| 64 | LDiT-B/1 | DINOv2-B | 1.2M | ✓ | (Noise, 0.4) | SDE | 6.79 |
| 64 | LDiT-B/1 | DINOv2-B | 1.2M | ✓ | (Random, 0.4) | SDE | 6.62 |
| 64 | LDiT-B/1 | DINOv2-B | 1.2M | ✓ | (Zero, 0.0) | SDE | 6.28 |
| 64 | LDiT-B/1 | DINOv2-B | 1.2M | ✓ | (Zero, 0.3) | SDE | 5.75 |
| 64 | LDiT-B/1 | DINOv2-B | 1.2M | ✓ | (Zero, 0.4) | ODE | 6.70 |
| 64 | LDiT-B/1 | CLIP | 1.2M | ✓ | (Zero, 0.4) | SDE | 10.81 |
| 80 | SiT-L/2 | DINOv2-B | 1.2M | ✓ | (Zero, 0.4) | SDE | 7.90 |

**Choice of visual encoder.**    To confirm that GMem is *encoder–agnostic*, we replace the default DINOv2-B feature extractor (Oquab et al., 2023) with the CLIP ViT-B/14 encoder (Radford et al., 2021). As in Yu et al. (2024), DINOv2-B remains the strongest option, but the CLIP variant still yields a competitive FID=10.81 on ImageNet 256, noticeably better than the vanilla LightningDiT-B/1 baseline (FID=15.82). These results indicate that GMem retains its benefit across representation sources.

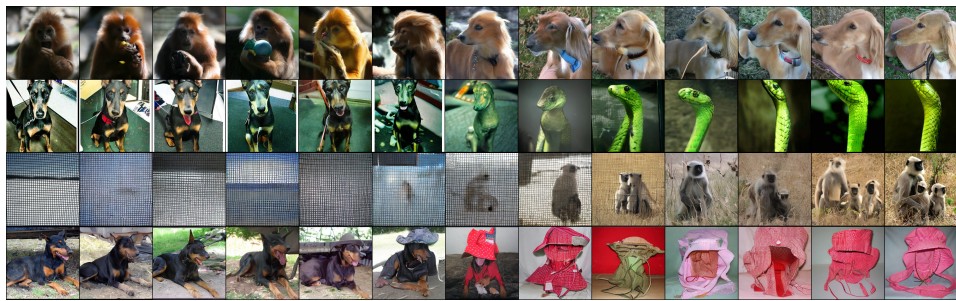

Figure 6: **Interpolation between memory snippets.** The first and last columns show the original memory snippets $s_1$ and $s_2$, respectively. The remaining columns show the generated images from the interpolated memory snippets $\hat{s}_i$.

**Larger architectures excel.** We examine the scalability of GMem by testing various model sizes and architectural configurations. Table 12 presents the FID scores of GMem across different model sizes on ImageNet $256 \times 256$: larger models not only converge faster but also achieve lower FID. This trend aligns with findings from Yu et al. (2024) and Ma et al. (2024) on diffusion transformers and extends to pixel-space generation.

**GMem generalize well with various networks and visual tokenizers.** We compare different network and visual tokenizers in Figure 12. LightningDiT consistently outperforms SiT under identical configurations, corroborating findings from Yao & Wang (2025). Additionally, DC-AE and VA-VAE tokenizer yields better results than SD-VAE, likely due to their larger parameter capacity.

**Masking strategy of memory bank.** We also explored two other masking strategies: *random mask* and *noise mask*. Specifically, *random mask* replaces a randomly selected portion of each batch with Gaussian noise, while *noise mask* adds noise to the entire memory snippet. The results for these two masking strategies are presented in Table 11. We found that zeroing out part of the snippet (the *Zero mask* strategy) consistently performed best across all experiments. Therefore, we adopted *Zero mask* for all major experiments.

**SDE solver is superior.** SDE solvers consistently outperform ODE solvers, reducing FID by $1.0$ (Table 11). Thus, SDE solvers are used in all main experiments.

### G.6 INTERPOLATION ON MEMORY SNIPPETS

In this section, we provide additional observations suggesting that the memory snippets used as input exhibit a degree of spatial smoothness, rather than degenerating into isolated point-to-point mappings as in a conventional autoencoder. Specifically, we demonstrate that GMem is capable of generating coherent and high-quality samples even when conditioned on interpolated memory snippets $\hat{s}$ that do not appear in the training set.

**Interpolation between memory snippets.** To assess this property, we perform an interpolation experiment on the ImageNet $256 \times 256$ using a model checkpoint trained for $140$ epochs (see Table 6 for details).

We randomly select two memory snippets $s_1$ and $s_2$ from the memory bank $\mathbf{M}$. We then create nine interpolated snippets $\hat{s}_i$ by linearly interpolating between $s_1$ and $s_2$ with interpolation coefficients $\alpha_i$ ranging from $0.1$ to $0.9$ in increments of $0.1$. The interpolated snippets are defined as:

$$\hat{s}_i = (1 - \alpha_i)s_1 + \alpha_i s_2, \quad \alpha_i = 0.1i, \quad i = 1, 2, \ldots, 9.$$

Each interpolated memory snippet $\hat{s}_i$ is then fed into the transformer block to generate images.

**Interpolation results.** The results of this interpolation experiment are presented in Figure 6 and Figure 7. We observe that the generated images from the interpolated memory snippets $\hat{s}_i$ are of high quality and exhibit smooth transitions between the two original memory snippets $s_1$ and $s_2$.

In the first row of Figure 6, we interpolate between an ape and a dog. The dog's face gradually transforms into a smoother visage, adapting to resemble the ape. This demonstrates representation

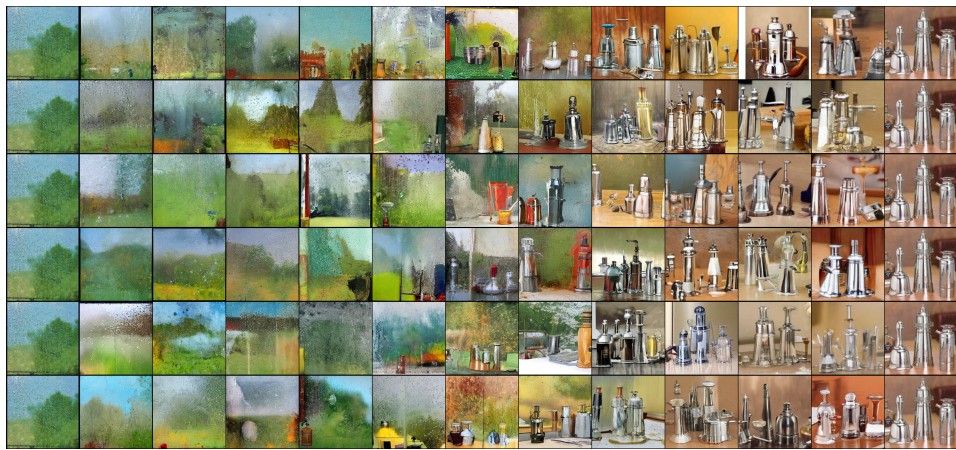

Figure 7: **A more elaborate interpolation experiment.** The first and last columns show the original memory snippets $s_1$ and $s_2$, respectively. The remaining columns show the generated images from the interpolated memory snippets $\hat{s}_i$. Different row stands for different noise applied when generating the images.

Table 12: (Left) **GMem consistently generates high-quality samples across different backbones and image tokenizers.** This table reports the FID of GMem with different backbones and visual tokenizers on ImageNet $256 \times 256$. For a fair comparison, we train all models for 64 epochs. ↓ means lower is better and all results reported are without classifier-free guidance. (Right) **CIFAR-10 generation performance (FID ↓).** All models trained at $32 \times 32$ resolution with 2048 epochs.

| Network | Visual tokenizer | Parameters (M) | Epoch | FID ($\downarrow$) |
|---|---|---|---|---|
| SiT-L/2 | SD-VAE | 458 | 64 | 6.49 |
| SiT-XL/2 | SD-VAE | 675 | 64 | 6.31 |
| LightningDiT-B/1 | VA-VAE | 130 | 64 | 5.70 |
| LightningDiT-B/1 | DC-AE | 130 | 64 | 5.97 |

| Model | Parameters (M) | Epoch | FID |
|---|---|---|---|
| SiT-S | 33 | 2048 | – |
| + REPA | 39.1 | 2048 | 11.04 |
| + GMem | 39.4 | 2048 | **5.12** |
| + GMem | 33.3 | 2048 | 9.43 |

space learned by GMem is semantically smooth. Surprisingly, in the second row, interpolating between a green snake and a long-faced dog results in a green reptilian creature that resembles both the snake and the dog. This indicates that when the model encounters unseen memory snippets, it can utilize the smooth latent space to generate images similar to those it has previously encountered.

The third and last rows showcase even more imaginative interpolations. Interpolating between a monkey and barbed wire results in an image of a monkey in a cage, while a dog and a red hat can be interpolated into a dog with a black gentleman's hat. These outcomes suggest that the similarities captured by the model are not limited to visual resemblance but also encompass more abstract semantic similarities in the latent space.

We believe that this semantic similarity arises because our memory bank introduces additional semantic information, enabling the model to better understand the content of images. Consequently, the model generates images that align more closely with human intuition, rather than merely memorizing the images corresponding to each snippet.

### G.7 Quantitative results on test-time domain adaptation

**Experimental setup.** We adopt the public `style_custom_dataset`[4] which provides six style domains absent from ImageNet: *watercolor, 3D, anime, flat-illustration, oil-painting*, and *sketch*. For each domain the training split contains 30 reference images paired with concise textual prompts describing their content (e.g. "a forest in watercolor").

**Evaluation metric.** Following Appendix G.7, we report LPIPS between every reference image and the model output that shares its prompt; a lower score indicates closer stylistic alignment and thus stronger test-time adaptation.

**Settings.** We compare three experimental settings :

---

[4]https://modelscope.cn/datasets/iic/style_custom_dataset/summary

Figure 8: **Fine-tuning to GMem.** Model adapts fast: by 20K fine-tuning steps, the model reaching the comparable FID=4.8 obtained when GMem is trained from scratch for 400K steps.

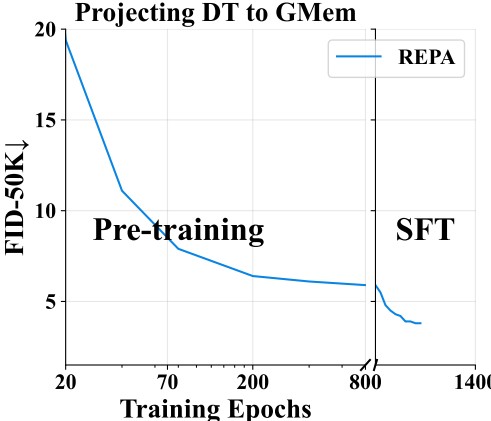

- **GMem** : insert a snippet extracted from each reference image into the external memory, then sample once per image and compute LPIPS with generated image and reference image.
- **SiT-XL+REPA** : LPIPS is reported as the smallest LPIPS value among all 50K images generated by REPA with the reference image.
- **ImageNet baseline** : for each reference we find its closest match (with smallest LPIPS) inside the ImageNet training set and record the LPIPS, representing an upper bound for a model that cannot adapt the new style.

**Results.** Table 13 summarises the outcome. Across all six domains, GMem (LightningDiT-XL+REPA+GMem) consistently attains the lowest LPIPS, outperforming both the REPA baseline and the ImageNet reference. The retrofitted model GMem already matches the GMem within 20K fine-tuning steps, confirming that memory adaptation capability is quickly recovered. These numbers demonstrate that external snippets enable genuine test-time domain adaptation while preserving sample diversity.

Table 13: **Test-time domain adaptation on six unseen styles** (LPIPS ↓, lower is better) at $256 \times 256$. GMem denotes LightningDiT-XL+REPA+GMem trained for 200 epochs without classifier-free guidance.

| Model | 3D | Anime | Flatillus. | Oil-paint | Sketch | Watercolor |
|---|---|---|---|---|---|---|
| ImageNet reference | 0.67 | 0.66 | 0.71 | 0.66 | 0.61 | 0.65 |
| SiT-XL+REPA$_{4M}$ | 0.69 | 0.68 | 0.73 | 0.68 | 0.67 | 0.67 |
| SiT-XL+REPA$_{4M}$+GMem | 0.65 | 0.63 | 0.67 | 0.59 | 0.61 | 0.60 |
| GMem | **0.64** | **0.62** | **0.67** | **0.60** | **0.56** | **0.61** |

## H    ADDITIONAL APPLICATIONS

In this section, we present experimental results demonstrating that the Memory Bank employed by GMem exhibits both cross-dataset transferability and adaptability to downstream T2I tasks.

### H.1    HIGH-RESOLUTION T2I GENERATION AND UNIFIED MULTIMODAL MODELS

To comprehensively evaluate GMem's capability at high-resolution generation, we additionally conduct experiment on high-resolution T2I generation. Together with Appendix G.4 , jointly demonstrate that GMem can maintain high fidelity when scaling to higher resolutions.

**Datasets.** All experiments are conducted on the same 23M image–text pairs used in OpenUni. For GMem, Stage 1 pretraining uses the image-only subset obtained by discarding textual annotations, while Stage 2 fine-tuning employs the full paired dataset.

**Evaluation metric.** We follow the official GenEval(Ghosh et al., 2023) protocol, which evaluates compositional T2I alignment at high resolution and uses it as the primary metric. The procedure strictly matches the OpenUni setting, including the prompt set, sampling strategy, and evaluation scripts.

**Experimental setup.** The unified multimodal architecture of OpenUni (3.6B parameters) is retained for both the baseline and GMem. For $1024 \times 1024$ generation, the patch size is doubled relative to the $512 \times 512$ configuration. GMem introduces an additional MLP adapter to perform text-to-snippet mapping, enabling end-to-end T2I generation without a memory bank.

The training schedules are as follows:

- **OpenUni baseline:** batch size $= 512$, 100K steps (2.226 epochs) on the full paired dataset.
- **GMem:** two-stage training procedure:
    1. *Stage 1 (image-only pretraining):* global batch size $= 64$, 32K steps (0.089 epochs).
    2. *Stage 2 (fine-tuning):* global batch size $= 64$, 3.2K steps (0.009 epochs) on image–text pairs.

Other hyperparameters, including optimizer, learning rate schedule, and sampling configuration, are kept identical to those in OpenUni (Wu et al., 2025), ensuring a fair comparison.

## H.2 DOWNSTREAM TASK ADAPTATION

We evaluate GMem's adaptability to novel domains by fine-tuning the ImageNet-pretrained model on three datasets that are different from natural images. In all cases, the pretrained GMem (650 epochs on ImageNet) is used as initialization. Fine-tuning is conducted for 20K steps per dataset, with hyperparameters (optimizer, learning rate schedule, and batch size) kept identical to those used in pretraining. For each target dataset, a new memory bank is extracted from the training data and employed during sampling.

**Datasets.** We select three representative benchmarks:

- **FFHQ** (Karras et al., 2019): a large-scale human face dataset designed for high-fidelity face synthesis.
- **MJHQ** (Li et al., 2024a): a dataset containing high-quality artistic and stylized images, challenging due to its distributional shift from natural photographs.
- **ACDC** (Bernard et al., 2018): a medical imaging dataset that is visually and semantically far from ImageNet, making it a stringent test for cross-domain adaptation.

Table 14: **ACDC adaptation with minimal fine-tuning.** We report FID ($\downarrow$) after Stage 1 memory-bank pretraining, with an optional short fine-tune (20K steps) on the target domain.

| Setting | Train (steps/epochs) | FID ($\downarrow$) |
|---|---|---|
| *ACDC (medical)* | | |
| GMem | 0 | 40.92 |
| GMem | 20K | 32.17 |
| SD 1.4 (Rombach et al., 2022) | – | 35.32 |

**Additional results.** Medical image generation adaptation: even though ACDC images were completely unseen during pre-training, 2h of fine-tuning lets GMem reach an FID=32.17 on par with an CogView baseline trained from scratch (FID=35.32).

## H.3 BANK-FREE T2I GENERATION

**Datasets.** We construct a high-quality subset of text–image pairs following the Micro-Diffusion (Sehwag et al., 2024). Specifically, we collect the top $1\%$ quality slice from CC-12M (Changpinyo et al., 2021), DiffusionDB (Wang et al., 2022b), and JDB (Sun et al., 2023), resulting in 203,592 pairs. All images are resized to $512 \times 512$ and center-cropped.

**Evaluation metric** We adopt **GenEval** (Ghosh et al., 2023) as the primary evaluation metric, strictly following the official protocol. No classifier-free guidance is used across all runs. As a secondary reference, we also report FID on MJHQ-30K.

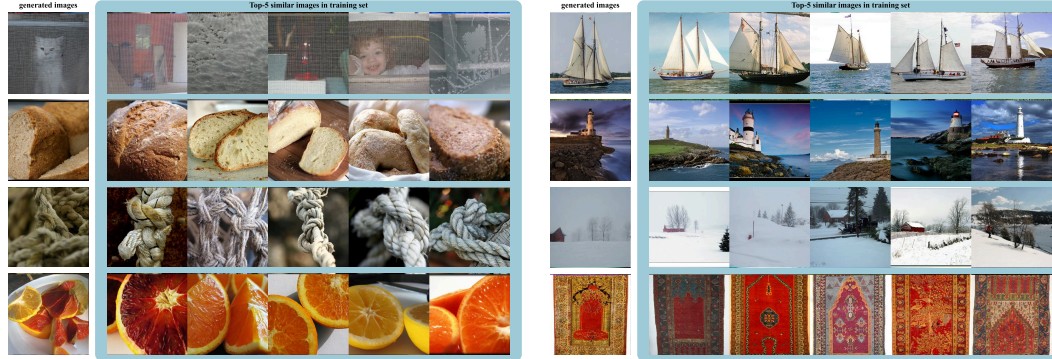

Figure 9: **Demonstration of diverse generation by GMem.** Selected samples from ImageNet $256 \times 256$ generated by the GMem. This figure demonstrates the diversity of images generated by GMem, which differ from the original training set in form, style, and color. This shows that GMem does not simply memorize images from the training set, but rather generates novel variations.

**Architecture:**  We add a lightweight two-layer MLP adapter on top of a pretrained Gemma-2b text encoder, enabling text-to-snippet mapping. Combined with GMem 's snippet-to-image pathway, this achieves end-to-end T2I generation without reliance on an external memory bank.

**Training procedure:**  Stage 1 pretraining uses 1.28M unlabeled images following the standard setup. Stage 2 fine-tuning uses 0.20M text–image pairs for 20K steps, with hyperparameters identical to pretraining. During both fine-tuning and inference, the memory bank is strictly disabled.

**Baselines.**  We study two variants to isolate where the T2I capability resides: (i) **GMem (full fine-tune)**: jointly fine-tune both the adapter and the diffusion network, (ii) **GMem (network frozen)**: fine-tune only the adapter, keeping the network fixed. We also compare against CLIP retrieval (a memory-based upper bound) and PixArt-$\alpha$ (a two-stage method without memory-based fine-tuning). Results are summarized in Table 15 .

Table 15: **Bank-free T2I generation.** GenEval $\uparrow$ is higher-better; FID $\downarrow$ is lower-better.

| Method | Total training (GPU-days) | Data size | MJHQ FID $\downarrow$ | GenEval $\uparrow$ |
|---|---|---|---|---|
| GMem (full fine-tune) | $75 + 3.5$ | 1.28M (unlabeled) + 0.2M | 7.36 | 0.52 |
| GMem (network frozen) | $75 + 3.5$ | 1.28M (unlabeled) + 0.2M | 10.34 | 0.32 |

**Results.**  Full fine-tuning substantially outperforms the frozen network variant (0.52 vs 0.32 GenEval), suggesting that compositional capability resides in the pretrained network rather than being solely attributable to the adapter. These results confirm that GMem can internalize knowledge into network parameters and achieve efficient bank-free T2I generation.

## H.4 TRANSFERABILITY OF THE MEMORY BANK

In this section, we demonstrate the transferability of the Memory Bank across different models. Specifically, we show that the Memory Bank can be transferred between GMem models trained on different datasets. While applying a Memory Bank extracted from low-resolution images to high-resolution models (e.g., Latent Diffusion Models) may result in decreased image sharpness due to information bottlenecks, it can still enhance the diversity of the generated image.

**Experimental setup.**  To investigate the transferability, we trained a Memory Bank $M_{\text{CIFAR}}$ on the CIFAR-10 dataset and directly transferred it to a model trained on ImageNet $256 \times 256$ GMem$_{\text{IN256}}$ to guide image generation. We used the checkpoint from the ImageNet model at 140 epochs for generation. The detailed experimental settings are provided in Table 5 and Table 6 .

**Method.**  Appendix H.4 demonstrates the transferability and generalization of the memory bank used to guide GMem across different datasets. Specifically, we train GMem$_{\text{IN256}}$ and GMem$_{\text{CIFAR}}$ models on ImageNet $256 \times 256$ and CIFAR-10, respectively, corresponding to memory banks of

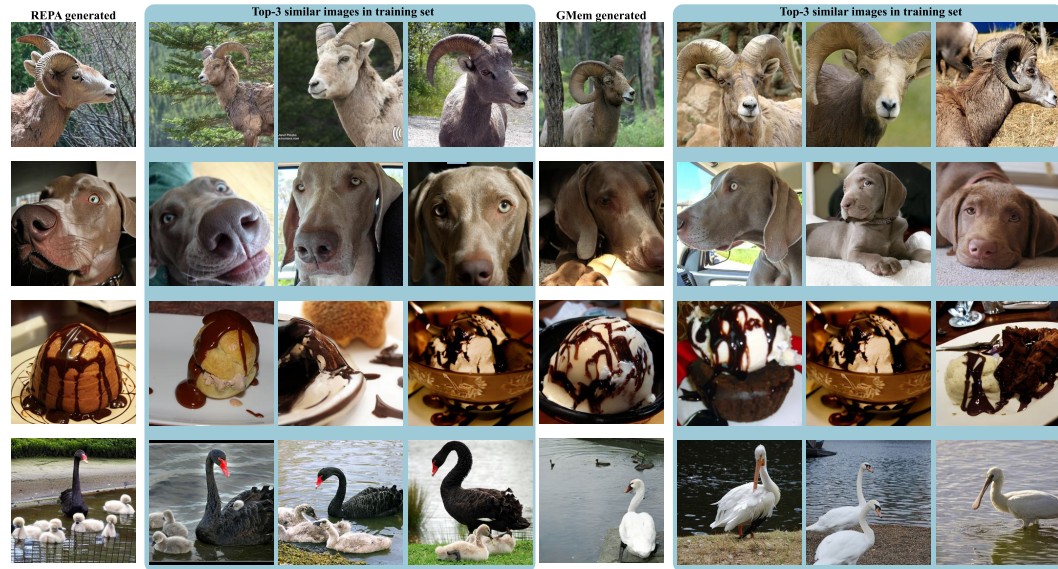

Figure 10: Qualitative comparison of per-class sample diversity on ImageNet. For each class, we show samples from the REPA baseline, and samples from GMem and nearest training images, Both REPA and GMem preserve the main class semantics while exhibiting diverse backgrounds, poses, and fine-grained details, yielding intra-class variability comparable to that of the training data.

$M_{\text{IN256}}$ and $M_{\text{CIFAR}}$. We then directly apply $M_{\text{IN256}}$ to guide the sampling process of GMem$_{\text{CIFAR}}$. Our results show that GMem$_{\text{CIFAR}}$ is still able to generate information consistent with the knowledge provided by $M_{\text{IN256}}$.

**Results**  Figure 11 presents the generation results of our method on the ImageNet $256 \times 256$ dataset. The images demonstrate that the transferred Memory Bank can effectively guide the high-resolution model. Though sharpness is limited due to information bottlenecks in memory snippets, it can improving the diversity of the generated images.

## I  DERIVATION OF DIFFUSION PROCESS

In this section, we provide a concise introduction to the training and sampling processes of flow-based and diffusion-based models. See Appendix I for more details.

Both diffusion-based (Ho et al., 2020; Dhariwal & Nichol, 2021) and flow-based models (Ma et al., 2024) derive their training procedures from a deterministic $T$-step noising process applied to the original data (Ma et al., 2024), formalized as:

$$\mathbf{x}_t = \alpha_t \mathbf{x}_0 + \sigma_t \boldsymbol{\epsilon}, \quad \boldsymbol{\epsilon} \sim \mathcal{N}(0, \mathbf{I}), \tag{5}$$

where $\mathbf{x}_t$ represents the noisy data at time $t$, $\mathbf{x}_0 \sim p(\mathbf{x})$ is a real data sample from the true distribution, $\alpha_t$ and $\sigma_t$ are time-dependent decreasing and increasing functions respectively satisfying $\alpha_t^2 + \sigma_t^2 = 1$.

As shown in (5), each marginal probability density $p_t(\mathbf{x}_t)$ represents the distribution of a Probability Flow Ordinary Differential Equation (Song et al., 2020b) (PF ODE). Its velocity field $\mathbf{v}(\mathbf{x}, t)$ is defined as:

$$\mathbf{v}(\mathbf{x}, t) = \dot{\alpha}_t \mathbb{E}[\mathbf{x}_0 \mid \mathbf{x}_t = \mathbf{x}] - \dot{\sigma}_t \mathbb{E}[\boldsymbol{\epsilon} \mid \mathbf{x}_t = \mathbf{x}], \tag{6}$$

where $\dot{\alpha}_t = \frac{\mathrm{d}\alpha_t}{\mathrm{d}t}$ and $\dot{\sigma}_t = \frac{\mathrm{d}\sigma_t}{\mathrm{d}t}$. Solving this ODE with initial condition $\mathbf{x}_T = \boldsymbol{\epsilon} \sim \mathcal{N}(0, \mathbf{I})$ yields the probability density function $p_0(\mathbf{x}_0)$, which approximate the ground-truth data distribution $p(\mathbf{x})$.

Alternatively, the aforementioned noise-adding process can be formalized as a Stochastic Differential Equation (Song et al., 2020b;a) (SDE):

$$\mathrm{d}\mathbf{x}_t = \boldsymbol{m}(\mathbf{x}_t, t)\, \mathrm{d}t + g(t)\, \mathrm{d}\mathbf{W}_t, \tag{7}$$

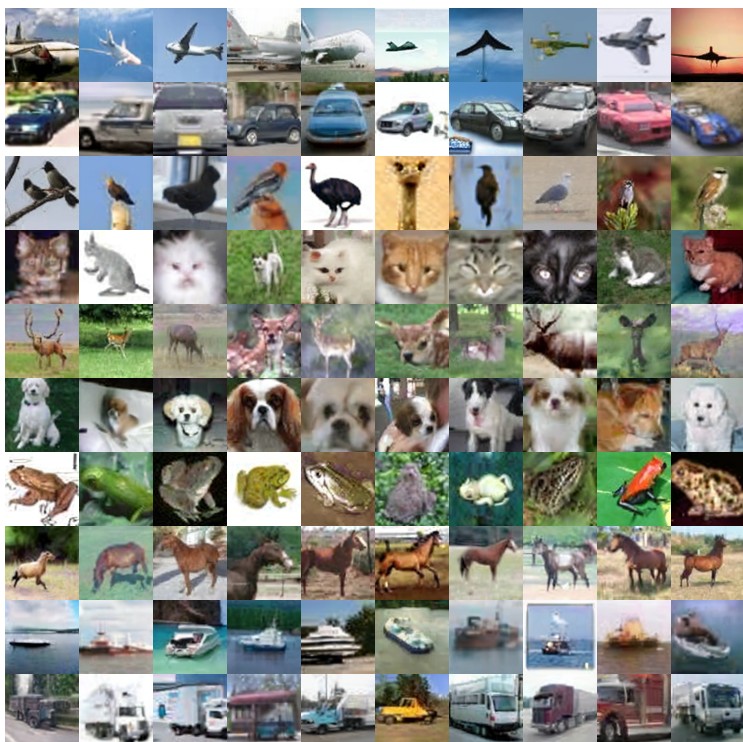

Figure 11: **Transferability of the memory bank.** Each row corresponding to a specifical class in CIFAR-10. Specifically, the class is from top to bottom: airplane, automobile, bird, cat, deer, dog, frog, horse, ship, and truck.

where $\mathbf{W}_t$ is a Wiener process (Hitsuda, 1968), $\boldsymbol{m}(\mathbf{x}_t, t)$ is the drift coefficient defined as $\boldsymbol{m}(\mathbf{x}_t, t) = -\frac{1}{2}\beta(t)\mathbf{x}_t$, and $g(t)$ is the diffusion coefficient, set as $g(t) = \sqrt{\beta(t)}$ with $\beta(t)$ being a time-dependent positive function controlling the noise schedule.

The corresponding reverse process is represented by the reverse-time SDE:

$$d\mathbf{x}_t = \left[\boldsymbol{m}(\mathbf{x}_t, t) - g(t)^2 s(\mathbf{x}_t, t)\right] dt + g(t)\, d\bar{\mathbf{W}}_t\,, \tag{8}$$

where $\bar{\mathbf{W}}_t$ is a reverse-time Wiener process, and $s(\mathbf{x}_t, t)$ is the score function, defined by the gradient of the log probability density:

$$s(\mathbf{x}_t, t) = \nabla_{\mathbf{x}_t} \log p_t(\mathbf{x}_t) = -\frac{1}{\sigma_t}\mathbb{E}\left[\boldsymbol{\epsilon} \mid \mathbf{x}_t = \mathbf{x}\right]\,. \tag{9}$$

By solving the reverse-time SDE in (8), starting from the initial state $\mathbf{x}_T = \boldsymbol{\epsilon} \sim \mathcal{N}(0, \mathbf{I})$, we can obtain $p_0(\mathbf{x}_0)$, thereby estimating the true data distribution $p(\mathbf{x})$.

### I.1 DENOISING DIFFUSION PROBABILISTIC MODELS

Diffusion models (Ho et al., 2020) aim to model a target distribution $p(\mathbf{x})$ by learning a gradual denoising process that transitions from a Gaussian distribution $\mathcal{N}(0, \mathbf{I})$ to $p(\mathbf{x})$. The core idea is to learn the reverse process $p(\mathbf{x}_{t-1}|\mathbf{x}_t)$ of a predefined forward process $q(\mathbf{x}_t|\mathbf{x}_0)$, which incrementally adds Gaussian noise to the data starting from $\mathbf{x}_0 \sim p(\mathbf{x})$ over $T$ time steps.

The forward process $q(\mathbf{x}_t|\mathbf{x}_{t-1})$ is defined as:

$$q(\mathbf{x}_t|\mathbf{x}_{t-1}) = \mathcal{N}\left(\mathbf{x}_t; \sqrt{1 - \beta_t}\, \mathbf{x}_0, \beta_t^2 \mathbf{I}\right)\,,$$

where $\beta_t \in (0, 1)$ are small, predefined hyperparameters.

In the DDPM framework introduced by Ho et al. (2020), the reverse process $p(\mathbf{x}_{t-1}|\mathbf{x}_t)$ is parameterized as:

$$p(\mathbf{x}_{t-1}|\mathbf{x}_t) =$$

$$\mathcal{N}\left(\mathbf{x}_{t-1}; \frac{1}{\sqrt{\alpha_t}}\left(\mathbf{x}_t - \frac{\beta_t}{\sqrt{1-\bar{\alpha}_t}}\,\varepsilon_\theta(\mathbf{x}_t, t)\right), \Sigma_\theta(\mathbf{x}_t, t)\right)$$

where $\alpha_t = 1 - \beta_t$, $\bar{\alpha}_t = \prod_{i=1}^{t} \alpha_i$, $\varepsilon_\theta(\mathbf{x}_t, t)$ is a neural network parameterized by $\theta$ and $\Sigma_\theta(\mathbf{x}_t, t)$ represents the learned variance.

The model is trained using a simple denoising autoencoder objective:

$$L_{\text{simple}} = \mathbb{E}_{\mathbf{x}_0, \varepsilon, t}\left[\|\varepsilon - \varepsilon_\theta(\mathbf{x}_t, t)\|_2^2\right],$$

where $\varepsilon$ is sampled from a standard normal distribution and $t$ is uniformly sampled from $\{1, \ldots, T\}$. For the variance $\Sigma_\theta(\mathbf{x}_t, t)$, Ho et al. (2020) initially set it to $\sigma_t^2 I$ with $\beta_t = \sigma_t^2$. However, Nichol & Dhariwal (2021) demonstrated that performance improves when $\Sigma_\theta(\mathbf{x}_t, t)$ is learned jointly with $\varepsilon_\theta(\mathbf{x}_t, t)$. They propose optimizing the variational lower bound (VLB) objective:

$$L_{\text{vlb}} = \exp\left(v \log \beta_t + (1-v) \log \tilde{\beta}_t\right),$$

where $v$ is a per-dimension component from the model output and $\tilde{\beta}_t = \frac{1-\bar{\alpha}_{t-1}}{1-\bar{\alpha}_t}\beta_t$.

By choosing a sufficiently large $T$ and an appropriate schedule for $\beta_t$, the distribution $p(\mathbf{x}_T)$ approaches an isotropic Gaussian. This allows for sample generation by starting from random noise and iteratively applying the learned reverse process $p(\mathbf{x}_{t-1}|\mathbf{x}_t)$ to obtain a data sample $\mathbf{x}_0$ (Ho et al., 2020).

### I.2 STOCHASTIC INTERPOLATING

In contrast to DDPM, flow-based models (Esser et al., 2024; Liu et al., 2023) address continuous time-dependent processes involving data samples $\mathbf{x}^* \sim p(\mathbf{x})$ and Gaussian noise $\varepsilon \sim \mathcal{N}(0, \mathbf{I})$ over the interval $t \in [0, 1]$. The process is formulated as:

$$\mathbf{x}_t = \alpha_t \mathbf{x}_0 + \sigma_t \varepsilon, \quad \text{with} \quad \alpha_0 = \sigma_1 = 1, \quad \alpha_1 = \sigma_0 = 0,$$

where $\alpha_t$ decreases and $\sigma_t$ increases as functions of $t$. There exists a probability flow ordinary differential equation (PF-ODE) characterized by a velocity field $\dot{\mathbf{x}}_t = \mathbf{v}(\mathbf{x}_t, t)$, ensuring that the distribution at time $t$ matches the marginal $p_t(\mathbf{x})$.

The velocity $\mathbf{v}(\mathbf{x}, t)$ is expressed as a combination of two conditional expectations:

$$\mathbf{v}(\mathbf{x}, t) = \mathbb{E}[\dot{\mathbf{x}}_t \mid \mathbf{x}_t = \mathbf{x}] = \dot{\alpha}_t\,\mathbb{E}[\mathbf{x}^* \mid \mathbf{x}_t = \mathbf{x}] + \dot{\sigma}_t\,\mathbb{E}[\varepsilon \mid \mathbf{x}_t = \mathbf{x}],$$

which can be approximated by a model $v_\theta(\mathbf{x}_t, t)$ through minimizing the training objective:

$$L_{\text{velocity}}(\theta) = \mathbb{E}_{\mathbf{x}^*, \varepsilon, t}\left[\|v_\theta(\mathbf{x}_t, t) - \dot{\alpha}_t \mathbf{x}^* - \dot{\sigma}_t \varepsilon\|^2\right].$$

This approach aligns with the reverse stochastic differential equation (SDE):

$$d\mathbf{x}_t = \mathbf{v}(\mathbf{x}_t, t)\,dt - \frac{1}{2}w_t s(\mathbf{x}_t, t)\,dt + \sqrt{w_t}\,d\bar{\mathbf{W}}_t,$$

where the score function $s(\mathbf{x}_t, t)$ is similarly defined as:

$$s(\mathbf{x}_t, t) = -\frac{1}{\sigma_t}\,\mathbb{E}[\varepsilon \mid \mathbf{x}_t = \mathbf{x}].$$

To approximate $s(\mathbf{x}_t, t)$, one can use a model $s_\theta(\mathbf{x}_t, t)$ with the training objective:

$$L_{\text{score}}(\theta) = \mathbb{E}_{\mathbf{x}^*, \varepsilon, t}\left[\|\sigma_t s_\theta(\mathbf{x}_t, t) + \varepsilon\|^2\right].$$

Since $s(\mathbf{x}, t)$ can be directly computed from $\mathbf{v}(\mathbf{x}, t)$ for $t > 0$ using the relation:

$$s(\mathbf{x}, t) = \frac{1}{\sigma_t} \cdot \frac{\alpha_t \mathbf{v}(\mathbf{x}, t) - \dot{\alpha}_t \mathbf{x}}{\dot{\alpha}_t \sigma_t - \alpha_t \dot{\sigma}_t},$$

it is sufficient to estimate either the velocity $\mathbf{v}(\mathbf{x}, t)$ or the score $s(\mathbf{x}, t)$.

According to Albergo et al. (2023), stochastic interpolants satisfy the following conditions when $\alpha_t$ and $\sigma_t$ are chosen such that: (i) $\alpha_t^2 + \sigma_t^2 > 0$ for all $t \in [0, 1]$, (ii) Both $\alpha_t$ and $\sigma_t$ are differentiable over the interval $[0, 1]$, (iii) Boundary conditions are met: $\alpha_1 = \sigma_0 = 0$ and $\alpha_0 = \sigma_1 = 1$.

These conditions ensure an unbiased interpolation between $\mathbf{x}_0$ and $\varepsilon$. Consequently, simple interpolants can be utilized by defining $\alpha_t$ and $\sigma_t$ as straightforward functions during training and inference. Examples include linear interpolants with $\alpha_t = 1 - t$ and $\sigma_t = t$, or variance-preserving (VP) interpolants with $\alpha_t = \cos\left(\frac{\pi}{2} t\right)$ and $\sigma_t = \sin\left(\frac{\pi}{2} t\right)$.

An additional advantage of stochastic interpolants is that the diffusion coefficient $w_t$ remains independent when training either the score or velocity models. This independence allows $w_t$ to be explicitly chosen after training during the sampling phase using the reverse SDE.

It's noteworthy that existing score-based diffusion models, including DDPM (Ho et al., 2020), can be interpreted within an SDE framework. Specifically, their forward diffusion processes can be viewed as predefined (discretized) forward SDEs that converge to an equilibrium distribution $\mathcal{N}(0, \mathbf{I})$ as $t \to \infty$. Training is conducted over $[0, T]$ with a sufficiently large $T$ (e.g., $T = 1000$) to ensure that $p(\mathbf{x}_T)$ approximates an isotropic Gaussian. Generation involves solving the corresponding reverse SDE, starting from random Gaussian noise $\mathbf{x}_T \sim \mathcal{N}(0, \mathbf{I})$. In this context, $\alpha_t$, $\sigma_t$, and the diffusion coefficient $w_t$ are implicitly defined by the forward diffusion process, potentially leading to a complex design space in score-based diffusion models (Karras et al., 2022).

## J   SOCIAL IMPACT

This paper presents work whose goal is to advance the field of Machine Learning. There are many potential societal consequences of our work, none which we feel must be specifically highlighted here.

## K   CODE

We release the code in supplementary material.

## L   LIMITATIONS

**Test-time domain adaptation.**   Although GMem enables on-the-fly style and concept transfer ( Figure D.1 ), adaptation can still fail when the inserted snippet lies far outside the training distribution. Preliminary evidence suggests that success correlates with visual and semantic proximity to seen data. A richer, modular memory organisation (Wu et al., 2022; Nichani et al., 2024; Mahdavi et al., 2023) may improve robustness, but we leave such design to future work.

**GMem → DT conversion.**   Our training-free reduction compresses a 1.28M-snippet bank to 1000 embeddings ( Section E.1 ). This aggressive shrinkage inevitably raises FID; exploring mixture or hierarchical embedding strategies that allocate several representatives per class could retain more information at the same memory budget. Developing principled criteria for selecting or distilling snippets is a promising research avenue.

**Scope of this work.**   Our goal is to introduce external, editable memory to diffusion transformers and to characterise the resulting trade-offs in efficiency, quality, and adaptability. Many implementation details—bank settings, advanced model conversion schemes, programmatic snippet editing—are deliberately kept simple and are left for future research to refine.

