# OpenReview forum: "Generative Modeling with Explicit Memory"
_ICLR.cc/2026/Conference — Submitted to ICLR 2026_

### Official Review · Reviewer_82a5 · 2025-10-17

**Soundness:** 3
**Presentation:** 3
**Contribution:** 3
**Rating:** 8
**Confidence:** 3

**Summary:**

This paper presents GMem, a method to improve diffusion image generation models by adding an external memory bank of semantic features (also referred to as memory snippets in the paper). These features are extracted during training, and, at inference time, are used as additional conditioning. The authors show that adding this external memory significantly speeds up training compared to a SiT and still reaches a good FID score. In addition, the memory bank can also be easily finetuned to adapt to new domains, and easily converted to help text-to-image generation.

**Strengths:**

-	Authors’ experiments show Gmem brings an order of magnitude speedup in training time without quality degradation. This is exciting.
-	GMem has the potential to become an easily-adaptable model in few-shot transfer learning. The memory snippet could be used for test-time editing.
-	The author discusses storage optimization techniques plus big-O analysis.
-	The authors tried multiple image encoders to show the generalizability of the method.
-	The idea is simple, thus practical to reproduce elsewhere.
-	The paper is well-structured and easy to follow. Discussion of related works is ample.

**Weaknesses:**

-	Using an external memory bank is novel, but there are similar works in RAG (retrieval-augmented generation), like RDM. Compared to them, the author’s innovation is in compressing the database, which is incremental.
-	Section 4.1 deserves more details. The authors say the memory bank is constructed by collecting N snippets from dataset D. But how does the author determine which N snippets? We can safely assume N << size(D), right? Or is N determined by a fixed ratio on top of the size of the data? This is not explained in the appendix either.
-	While the authors apply masking techniques to avoid over-memorization (the model just decodes a snippet), the metrics in the experiments cannot measure the true diversity of the output. It is recommended to compare the similarity with the training set images to see if it has over-memorization.
-	In Section 4.3, the reviewer does not fully understand why mapping a sampled noise to some element in the bank ensures good generation diversity. Why is a normal distribution not a uniform distribution used? Would this make the center indices more likely to be chosen?
-	In Fig.3 right side, the arrows between “class embed”, “Memory bank”, and “Text encoder” are a bit confusing. At first sight, readers might think there is some data transformation between them and not realize that these refer to different models.

**Questions:**

-	Line 693 and 696 have duplicated reference.
-	Line 514, the cited reference seems wrong. Should be Retrieval-Augmented Diffusion Models by Blattmann et al.

---

> ### Author Response · Authors · 2025-11-26
> **Response to Reviewer 82a5(1/3)**
>
> We sincerely thank you for your positive assessment of GMem’s practical impact, adaptability, and clarity of presentation, and we truly appreciate the recognition of our efforts on efficiency, generalizability, and reproducibility. We would also like to kindly clarify a few minor points as follows:
>
> > W1: the author’s innovation is in compressing the database, which is incremental.
>
> **Core Innovations.** We appreciate the reviewer’s detailed review and we would like to kindly clarify that our fundamental contribution lies in **establishing GMem as a unified framework that bridges the gap between sparse label-based (class-conditional) generation and dense conditional text-to-image (T2I) synthesis.** Building on this unified view, we use an explicit semantic memory to control conditioning strength and derive a simple, unified pathway that smoothly moves from low-strength image-only pretraining to high-strength T2I fine-tuning.
>
> Concretely, our core innovation can be divided into three aspects:
>
> 1. **Conditioning view of conditional diffusion models.** We analyze conditional diffusion models from the viewpoint of conditioning strength, and show that class-conditional diffusion, GMem, and T2I diffusion can be placed on a single continuum (Section 3). Class-conditional models use a coarse global class label as a very weak conditioning signal, while T2I models rely on dense text latents as a strong, fine-grained conditioning signal. GMem, driven by image-derived semantic snippets, lies in the middle and thus serves as a bridge that connects the two extremes within a single conditioning framework.
> 2. **Explicit semantic memory for conditioning control.** Within this framework, we take memory snippets as the controllable conditioning signal—compact semantic descriptors extracted from images that capture their key content and structure (Section 4.1). By editing snippets, compressing them into prototypes, or reducing their number, we can systematically manipulate the memory bank and thereby flexibly adjust the effective conditioning strength within a single architecture (Section 5.2). In this formulation, we further show that traditional class-conditional diffusion models emerge as a special case of GMem, where the memory bank for each class collapses into a single prototype embedding (Appendix E), and that the resulting memory representations transfer across datasets (Appendix H.4).
> 3. **Unified weak-to-strong conditioning (image-only → T2I) pathway.** This conditioning-strength view directly leads to a unified training pathway: we start from low-strength conditioning by pretraining on image-only data with snippet-based conditioning (Sections 5.2), and then move towards high-strength conditioning by transitioning to text-latent–driven conditioning for T2I (Section 4.4). In other words, we stay within the same generative framework and gradually shift the source and strength of conditioning—from visual snippets to text latents—rather than training separate models for image-only generation and T2I.
>
> In practice, this brings two concrete advantages:
>
> 1. **Faster training convergence:** The rich semantic information within explicit memory rectifies flow trajectories, directly enabling a $50\times$ training acceleration on ImageNet compared to SiT.
> 2. **A data- and compute-efficient T2I generation pipeline.** The unified framework lowers the barrier for high-quality generation, achieving SoTA T2I results with significantly reduced data and compute resources compared to training from scratch.
>
> Together, GMem establishes explicit memory not just as an acceleration technique, but as a rigorous, scalable paradigm that unifies class-conditional tasks and text-to-image generation, while enabling efficient training and flexible downstream adaptation.

---

> ### Author Response · Authors · 2025-11-26
> **Response to Reviewer 82a5(2/3)**
>
> > W2: how does the author determine which N snippets
>
> We thank the reviewer for pointing out that the choice of the memory bank size ($N$) was not clearly explained, and we apologize for the missing information.
>
> **Size of memory bank.** It is worth noting that the usage the memory bank is setting-dependent: it is explicit and instance-level for moderate-scale class-conditional benchmarks, but becomes fully bank-free in our large-scale T2I setup.
>
> - **Memory bank in class-conditional generation (ImageNet).** For class-conditional generation on standard benchmarks, we simply use one snippet per training image, which yields the best result reported in Table 1. As noted in Section 4.3, the storage cost for a $1.28\text{M}$-entry bank is approximately $3$ GB, which is computationally manageable.
> - **Bank-free in large-scale T2I generation.** For datasets with $\ge 10\text{M}$ images (e.g., the $23\text{M}$ pairs in OpenUni), a traditional retrieval-based memory bank becomes prohibitively expensive. Therefore, in this regime, we remove the memory bank and instead adopt the Bank-Free T2S adapter described in Section 4.4, which internalizes the memory into the model parameters.
>
> **Effect of different bank sizes.**
> Empirically, GMem is highly robust to reductions in $N$. As detailed in Table 11 and Appendix F.8, we performed a memory-capacity sweep on ImageNet ($256\times256$), varying $N$ from $1.28\text{M}$ down to $1\text{K}$ entries.
>
> - **Diminishing Returns.** Larger banks improve FID monotonically, but with diminishing returns. As shown in Table 11 (Appendix F.8), halving the bank size only worsens FID by $\sim0.08\%$ on average, and even a $10\times$ reduction in $N$ (from $1.28\text{M}$ to $128\text{K}$ snippets) results in merely a $0.25\%$ relative increase in FID.
> - **Minimal Degradation.** Overall, even a $10\times$ reduction in bank size results in only a $0.25\%$ relative degradation in FID, indicating that GMem is highly robust to aggressive memory compression.
>
> We summarize the impact of $N$ on ImageNet in the table below:
>
> | **Dataset**  | **Bank size N** | **Relative size (vs full)** | **FID** |
> | ------------ | --------------- | --------------------------- | ------- |
> | ImageNet 256 | 1.28M           | 1.0 $\times$                | 5.70    |
> | ImageNet 256 | 128K            | 0.1 $\times$                | 5.72    |
> | ImageNet 256 | 64K             | 0.05 $\times$               | 5.85    |
>
> > W3: recommended to compare the similarity with the training set images to see if it has over-memorization
>
> We appreciate the reviewer's recommendation to rigorously verify that the use of explicit memory snippets does not lead to over-memorization. We have addressed this concern through both quantitative metrics and direct visual comparisons with the training set:
>
> **(i) Quantitative evidence (Appendix G.2).** As detailed in Appendix G.2, we explicitly measure the similarity between generated samples and their nearest neighbors in the training set using LPIPS, SSIM, and PSNR. GMem matches the diversity metrics of the strong SiT+REPA baseline under matched FID, and differs from VAE baselines.
>
> **(ii) Qualitative evidence (Figures 4 and 8).** We provide visual evidence that the model synthesizes novel content rather than retrieving stored data:
>
> - **Nearest-neighbor comparison (Figure 8).** We visualize generated samples alongside their top-$5$ nearest neighbors from the training set. The comparison highlights significant differences in form, style, and color, demonstrating that GMem generates novel variations.
> - **Novel concept generalization (Figure 4).** We show that GMem can also adapt to novel images and concepts (e.g., interpolations between “dog” and “hat”) that do not exist in the training set.
>
> **(iii) New visualization compared with REPA. (Figure 10)** To further address the reviewer’s concern, we have added a direct **side-by-side visualization** comparing “REPA generated”, “GMem generated”, and “Training set (nearest neighbors)”. This comparison visually confirms that GMem maintains the same level of generative diversity as the baseline (REPA) while remaining distinct from the training data.
>
> Taken together, we believe that the consistent quantitative metrics (LPIPS/SSIM/PSNR), combined with the visual nearest-neighbor analysis, provide robust evidence to alleviate concerns regarding sample diversity and over-memorization.

---

> ### Author Response · Authors · 2025-11-26
> **Response to Reviewer 82a5(3/3)**
>
> > W4. why mapping a sampled noise to some element in the bank ensures good generation diversity. Why is a normal distribution not a uniform distribution used … Would this make the center indices more likely to be chosen?
>
> We appreciate the reviewer’s invaluable suggestion.
>
> **Diversity source.** We apologize for the confusion and clarify that we do not intent to claim that “mapping a sampled noise to some element in the bank” guarantees good generation diversity. Diversity arises because GMem provides a conditioning signal that lies between sparse class labels and dense text-to-image mappings, retaining sufficient entropy to preserve the data diversity. Furthermore, our random masking strategy prevents the model from degenerating into a rigid one-to-one mapping, ensuring that it learns a more general and diverse distribution (as described in Section 3).
>
> **Uniform snippet selection during training.** In Section 4.3, the mapping from the sampled noise to a snippet index is implemented via a normal-to-uniform transform, so that each snippet is still selected with uniform probability—there is no bias toward “center indices”.
>
> This design primarily serves to maintain consistency with standard diffusion modeling interfaces (i.e., initializing with Gaussian noise) and to facilitate a two-way conversion between conventional class-conditional DiTs and GMem (as described in Appendix E).
>
> > W5. readers might think there is some data transformation between them and not realize that these refer to different models
>
> We appreciate the reviewer’s valuable suggestion and apologize for the confusion. We have revised the figure labels and caption to explicitly state that the components are substitutable modules, ensuring readers do not interpret this as a data transformation pipeline.
>
> Crucially, this figure is designed to illustrate GMem's core innovation: acting as a unified framework that bridges the gap between sparse class-conditional generation and dense text-to-image (T2I) synthesis. The arrows represent a conceptual continuum along which GMem can structurally "degrade" into a standard DiT (by collapsing the bank into class embeddings) or "upgrade" to a T2I model (by replacing the bank with a text encoder).
>
> > Q1: Line 693 and 696 have duplicated reference.  Q2: Line 514, the cited reference seems wrong. Should be Retrieval-Augmented Diffusion Models by Blattmann et al.
>
> We thank the reviewer for carefully checking the references, and we have corrected both issues accordingly:
>
> - Line 693 & 696. We have removed the duplicated entries for Rombach et al. (2022) in the bibliography.
> - Line 514. We have corrected the citation to refer to *Retrieval-Augmented Diffusion Models* by Blattmann et al. as suggested.

---

> > ### Comment · Reviewer_82a5 · 2025-11-28
> >
> > Thanks to the authors for the answers. My concerns are addressed.

---

### Official Review · Reviewer_7j2m · 2025-10-23

**Soundness:** 3
**Presentation:** 3
**Contribution:** 2
**Rating:** 4
**Confidence:** 2

**Summary:**

The paper proposes GMem, a diffusion-based generative framework that conditions image synthesis on high-quality semantic representations stored in an external memory bank instead of noisy text prompts. This design enables dramatically faster training, which achieves an SOTA FID of 1.53 on the ImageNet dataset, 50× faster than SiT. It also enjoys rapid downstream adaptation to new domains.

**Strengths:**

1. The paper is well written. Figures (especially Figure 3) effectively illustrate the architecture, training/sampling pipelines, and relationships to prior work.
2. The core idea is conceptually fresh and thoughtfully motivated.
3. The paper focuses on three timely and important challenges in generative modeling: training inefficiency, poor adaptability, and reliance on large paired datasets.

**Weaknesses:**

1. GMem’s core innovation relies on storing a full-dataset memory bank (e.g., 1.28M snippets for ImageNet), the cost is manageable for ImageNet but prohibitive for LAION-scale T2I.
2. Section 5.3 states that an ImageNet-pretrained GMem can adapt to new domains (anime, faces) in ~20K fine-tuning steps. However, there is no comparison to fine-tuning a strong baseline model on the same downstream tasks.

**Questions:**

See Weaknesses.

---

> ### Author Response · Authors · 2025-11-26
> **Response to Reviewer 7j2m(1/2)**
>
> We sincerely thank the reviewer for the constructive and detailed feedback, and we greatly appreciate the recognition of our clear presentation, the conceptual freshness of our core idea, and our focus on training and data efficiency, adaptability in generative modeling.
>
> > W1: Core innovation relies on storing a full-dataset memory bank
>
> We appreciate the reviewer’s thoughtful analysis. We agree that explicitly storing a memory bank for LAION-scale datasets would be prohibitive, which is a well-known limitation of traditional Retrieval-Augmented Diffusion methods. **However, our core innovation does *not* rely on storing a full-dataset memory bank.** Instead, GMem is designed to overcome this limitation via its Bank-Free design, and we re-emphasize these aspects below.
>
> **Clarification: Manageable cost and Bank-Free Design.**
>
> - **For Class-Conditional Generation:** Extracting the full ImageNet memory bank (1.28M entries) requires only ~3 GB of storage. Furthermore, in Section 4.3 and Appendix C, we introduce SVD-based decomposition strategies to reduce this storage further.
> - **For Large-Scale T2I:** We explicitly adopt the Bank-Free T2S design described in Section 4.4. By training a lightweight adapter to "internalize" memory into model parameters, GMem eliminates the need for an external memory bank during T2I inference. This effectively resolves the storage bottlenecks associated with scaling to massive datasets (e.g., LAION).
>
> **Core Innovations.** Our fundamental contribution lies in **establishing GMem as a unified framework that bridges the gap between sparse label-based (class-conditional) generation and dense conditional text-to-image (T2I) synthesis.** Building on this unified view, we use an explicit semantic memory to control conditioning strength and derive a simple, unified pathway that smoothly moves from low-strength image-only pretraining to high-strength T2I fine-tuning (Section 3).

---

> ### Author Response · Authors · 2025-11-26
> **Response to Reviewer 7j2m(2/2)**
>
> > W2: no comparison to fine-tuning a strong baseline model on the same downstream tasks
>
> We appreciate the reviewer’s valuable insight. We agree that a direct "relative" comparison against strong baselines such as REPA is essential, and we provide such an analysis below.
>
> **Conceptual advantage: GMem is an "open-set" generative model.** First, we highlight a fundamental structural difference: conventional class-conditional models (such as DiT and SiT) are essentially "closed" with respect to their label space—incorporating a new domain typically requires modifying the architecture (appending new class embeddings) and retraining them to map new labels to visual features. In contrast, GMem is "open": it conditions on semantic snippets not tied to a fixed vocabulary. This allows GMem to adapt to out-of-domain data immediately via the memory bank without architecture changes, offering a structural efficiency advantage that standard baselines lack.
>
> **Empirical relative comparison.** To rigorously quantify this advantage, we implemented a "relative" comparison on the FFHQ dataset using the LightningDiT-XL backbone. We compared GMem against two variants of REPA designed to handle label-space extension:
>
> - **REPA (new-class random init):** We append a new learnable class embedding $c_{\text{new}}$ initialized with a standard normal distribution. All FFHQ images are assigned to this label, and we fine-tune both the embedding and backbone.
> - **REPA (new-class avg-emb init):** To better evaluate knowledge transfer, we initialize $c_{\text{new}}$ as the average of all pre-trained class embeddings (excluding the null embedding). This allows the baseline to leverage the "average" semantic knowledge of the pre-trained model.
>
> Both GMem and the REPA baselines were fine-tuned under identical optimization settings. The results are shown in the table below:
>
> | Method              | Training speed | 0 (zero-shot) | 7 epochs | 180 Epochs |
> | ------------------- | -------------- | ------------- | -------- | ---------- |
> | GMem                | 1.55 s/step    | 8.65          | 8.55     | 7.08       |
> | REPA (random init)  | 1.62 s/step    | -             | 96.15    | 27.55      |
> | REPA (avg-emb init) | 1.62 s/step    | -             | 74.28    | 21.32      |
>
> **Benefit of initialization.** Comparing the two REPA variants, REPA (avg-emb init) indeed accelerates convergence (FID $74.28$ vs $96.15$ at $7$ epochs), confirming that utilizing prior knowledge helps.
>
> **Superiority of GMem.** However, GMem radically outperforms even REPA (avg-emb init).
>
> - **Zero-shot adaptation.** GMem achieves a strong FID of $8.65$ immediately without any gradient updates, whereas the REPA baselines cannot generate valid samples for the new class in the zero-shot setting.
> - **Adaptation speed.** With only $7$ epochs, GMem maintains high image quality (FID $8.55$), whereas REPA (avg-emb init) yields poor results (FID $74.28$). This substantial performance advantage persists even after prolonged training ($180$ epochs).
>
> We argue that these results empirically validate GMem as a general-purpose base model tailored for rapid downstream adaptation, substantiating one of the primary contributions of this work.

---

> > ### Comment · Reviewer_7j2m · 2025-11-28
> >
> > I thank the authors for addressing the raised issues. Your clarifications have strengthened my understanding of the rationale behind the paper. I will revise my score to 6.

---

### Official Review · Reviewer_gjUM · 2025-10-25

**Soundness:** 3
**Presentation:** 4
**Contribution:** 2
**Rating:** 4
**Confidence:** 3

**Summary:**

This paper introduces GMem (Generative Modeling with Explicit Memory), a diffusion-based generative framework that conditions
generation on semantic information, which stored in an external memory bank, extracted directly from the data. GMem significantly improved training efficiency. Moreover, the authors also extend GMem to a text-to-image (T2I) setup with strong data and time efficiency.

**Strengths:**

- The paper is well written and easy to follow.
- GMem demonstrates comprehensive empirical validation - the experiments are broad, spanning CIFAR-10, ImageNet (256×256, 512×512), and T2I tasks. And it also shows strong emperical resultls - outperforming prior baselines such as SiT, REPA, and PixelArt-α on various metrics covering efficiency and quality.

**Weaknesses:**

- Limited conceptual novelty. The paper does not sufficiently clarify what is fundamentally new. The core idea is conceptually close to retrieval-augmented diffusion (e.g., REPA, KNN-Diffusion, Memory-Driven T2I).
- Lack of theoretical or mechanistic insight. The paper does not analyze why explicit memory accelerates convergence or improves sample quality.
- Extracting features from a pretrained model to build the memory bank requires running inference on millions of images, which can take hundreds of GPU-hours for ImageNet-scale datasets, but this cost is probably not included in training time.
- Training converges faster because the network conditions on semantically rich embeddings rather than noisy text or random class tokens, experical results are very strong but it's not a new algorithmic insight.

**Questions:**

- The reported “50×” training speedup, is it due to property of GMem? Is the speedups measured under identical optimization and data settings as baselines?

---

> ### Author Response · Authors · 2025-11-26
> **Response to Reviewer gjUM(1/3)**
>
> We sincerely thank the reviewer for the constructive and detailed feedback. We appreciate the acknowledgment of our presentation and experimental comprehensiveness.
>
> > W1: Limited **conceptual** novelty. The paper does not sufficiently clarify what is fundamentally new.
>
> We appreciate the reviewer’s valuable insight, and we take this opportunity to clarify the core innovations of GMem.
>
> **Core Innovations.** Our fundamental contribution lies in **establishing GMem as a unified framework that bridges the gap between sparse label-based (class-conditional) generation and dense conditional text-to-image (T2I) synthesis.** Building on this unified view, we use an explicit semantic memory to control conditioning strength and derive a simple, unified pathway that smoothly moves from low-strength image-only pretraining to high-strength T2I fine-tuning.
>
> Concretely, our core innovation can be divided into three aspects:
>
> 1. **Conditioning view of conditional diffusion models.** We analyze conditional diffusion models from the viewpoint of conditioning strength, and show that class-conditional diffusion, GMem, and T2I diffusion can be placed on a single continuum (Section 3). Class-conditional models use a coarse global class label as a very weak conditioning signal, while T2I models rely on dense text latents as a strong, fine-grained conditioning signal. GMem, driven by image-derived semantic snippets, lies in the middle and thus serves as a bridge that connects the two extremes within a single conditioning framework.
> 2. **Explicit semantic memory for conditioning control.** Within this framework, we take memory snippets as the controllable conditioning signal—compact semantic descriptors extracted from images that capture their key content and structure (Section 4.1). By editing snippets, compressing them into prototypes, or reducing their number, we can systematically manipulate the memory bank and thereby flexibly adjust the effective conditioning strength within a single architecture (Section 5.2). In this formulation, we further show that traditional class-conditional diffusion models emerge as a special case of GMem, where the memory bank for each class collapses into a single prototype embedding (Appendix E), and that the resulting memory representations transfer across datasets (Appendix H.4).
> 3. **Unified weak-to-strong conditioning (image-only → T2I) pathway.** This conditioning-strength view directly leads to a unified training pathway: we start from low-strength conditioning by pretraining on image-only data with snippet-based conditioning (Sections 5.2), and then move towards high-strength conditioning by transitioning to text-latent–driven conditioning for T2I (Section 4.4). In other words, we stay within the same generative framework and gradually shift the source and strength of conditioning—from visual snippets to text latents—rather than training separate models for image-only generation and T2I.

---

> ### Author Response · Authors · 2025-11-26
> **Response to Reviewer gjUM(2/3)**
>
> > W2: Lack of theoretical or mechanistic insight.
>
> We appreciate the reviewer’s valuable insight. In the revised version (Section 3), we provide a mechanistic explanation for the observed acceleration. We identify the main diffusion training bottleneck as the geometric complexity of the probability-flow vector field, which is tied to conditioning strength. Typically, strong, informative conditioning drives the dynamics toward simple, almost straight trajectories between the noisy inputs and the underlying data modes, whereas standard class-label conditioning is too weak and coarse, leading to highly curved, oscillatory trajectories that are hard to learn.
>
> GMem addresses this by utilizing memory snippets to provide a more informative conditioning signal, thereby rectifying the probability-flow trajectories into shorter, straighter, and non-oscillatory paths and simplifying the learning objective.
>
> Our visualization on a 1D toy example (Figure 3 in the revised paper) empirically confirms this geometric simplification. While standard class-conditional baselines exhibit oscillating, highly curved trajectories—driven by the conflict between early-stage mean-seeking and late-stage mode-seeking dynamics—in contrast, GMem produces straight, non-oscillating paths.
>
> > W3: extracting features from a pretrained model to build the memory bank requires running inference on millions of images,
>
> We thank the reviewer for considering the practical cost of feature extraction. We clarify that this overhead is minor.
>
> **One-time preprocessing cost:** As shown in the table below, processing and saving ImageNet ($1.28\text{M}$ images) takes approximately $11.5$ GPU-hours on a standard A100. In terms of end-to-end comparisons, the SiT-XL baseline requires approximately $17,000$ GPU-hours, whereas GMem requires only $\sim400$ total GPU-hours to get comparable FID. This represents a $\sim 42.5\times$ speedup, thereby rendering the $11.5$ GPU-hours of feature extraction cost negligible.
>
> | **Dataset**  | **Size** | **Extraction Time (A100 hours)** | Training/Fine-tuning Time (A100 hours) | FID   |
> | ------------ | -------- | -------------------------------- | -------------------------------------- | ----- |
> | ImageNet-256 | 1.28M    | 11.5                             | 384                                    | 1.54  |
> | FFHQ         | 70K      | 0.6                              | 38.4                                   | 6.62  |
> | MJHQ         | 30K      | 0.3                              | 38.4                                   | 6.10  |
> | ACDC         | 4K       | 0.03                             | 38.4                                   | 32.17 |

---

> ### Author Response · Authors · 2025-11-26
> **Response to Reviewer gjUM(3/3)**
>
> > W4: experical results are very strong but it's not a new algorithmic insight
>
> We thank the reviewer for recognizing the strength of our empirical results. We respectfully argue that the core value of GMem goes beyond providing a stronger condition; rather, the use of explicit memory snippets fundamentally transforms how the model learns and transfers knowledge. We highlight two key algorithmic insights:
>
> **Traditional DiT is a compressed GMem.** We kindly provide a new perspective that standard class-conditional diffusion models (e.g., DiT) are essentially a degenerate case of GMem where the memory bank is compressed into a static lookup table (i.e., one fixed embedding per class). Our "Dual Conversion" experiments (Appendix E) empirically validate this equivalence.
> We argue that this insight offers significant community value by enabling model upgrading. We show that widely available, open-source checkpoints (e.g., SiT) can be "upgraded" to GMem with minimal fine-tuning (e.g., ~20K steps), matching the performance of training from scratch. This lowers the barrier for the community to adopt explicit memory without expensive retraining.
>
> **Decoupled T2I Generation Pipeline.** GMem offers a new perspective by decoupling the complex Text-to-Image task into two distinct sub-problems: Snippet-to-Image (S2I) a with snippet-based conditioning and Text-to-Snippet (T2S) for text latent conditioning. This separation provides two main benefits:
>
> - **Reusability of the S2I Backbone.** The S2I module serves as a robust, general-purpose base model that can be directly reused without retraining the generative network from scratch. A similar efficiency pattern is observed in domain adaptation: the pre-trained S2I backbone adapts to visually distinct domains—such as Anime (MJHQ) and Medical Imaging (ACDC)—with just ~20K fine-tuning steps, matching the quality of models trained from scratch.
>
> - **Efficient Training of T2S Alignment.** The T2S stage is simplified to a lightweight feature alignment task. This unlocks substantial data and compute efficiency: as shown in Table 3, this paradigm allows GMem to match the performance of PixArt-$\alpha$ using only $1/17$ of the data and $1/9$ of the training time.
>
>
>
> > Q1: The reported “50×” training speedup, is it due to property of GMem?
>
> We appreciate the reviewer’s detailed review. We confirm that the speedup is measured under **identical optimization and data settings** as the baselines, ensuring a fair comparison.
>
> **Experimental Settings:**
>
> - SiT Backbone: we strictly follow the SiT/REPA protocol: AdamW, learning rate $1\times10^{-4}$, and batch size 256 (see Table 7).
> - LightningDiT Backbone: we utilize the official LightningDiT configuration: AdamW, learning rate $2\times10^{-4}$, and batch size 1024 (see Table 6).
> - Metric: to account for batch size differences across baselines, we uniformly use the **number of training epochs** (i.e., passes over the dataset) as our efficiency metric.
>
> We attribute this efficiency to the informative guidance provided by memory snippets, which effectively rectifies the probability-flow trajectories (as discussed in Section 3). This rectification simplifies the target velocity field, significantly reducing the optimization difficulty and resulting in the observed training acceleration. We have included a detailed theoretical analysis and numerical simulations in the revised version to support this claim.

---

### Official Review · Reviewer_Rz5N · 2025-11-11

**Soundness:** 3
**Presentation:** 2
**Contribution:** 2
**Rating:** 4
**Confidence:** 3

**Summary:**

This paper tackles the problem of inefficient training issues in diffusion models, which often rely on large-scale text–image pairs for supervision. The authors propose Generative Modeling with Explicit Memory (GMem), a framework that conditions generation on high-quality semantic features extracted directly from data and stored in an external memory bank. This memory-based conditioning provides a more accurate and efficient guidance signal, enabling up to 50× faster training and state-of-the-art performance on ImageNet. Moreover, GMem allows rapid domain adaptation and a data-efficient text-to-image pipeline, reducing both dataset size and computational cost while maintaining competitive quality.

**Strengths:**

1. This paper addresses an important and timely problem: improving the efficiency of training generative models.

2. The proposed methods are conceptually sound, clearly presented, and relatively easy to follow.

3. The paper presents a comprehensive set of experiments, demonstrating the proposed method’s effectiveness across multiple dimensions, including generation quality, adaptability to new tasks, and training efficiency.

**Weaknesses:**

1. The dependency on REPA appears to be a crucial factor that requires further clarification. In Table 1, most of GMem’s results are reported on top of REPA, making it unclear whether GMem alone yields significant improvements. A comparison isolating GMem’s contribution would strengthen the claim of its independent effectiveness.

2. The claimed advantages of the Bank-Free T2I design (lines 307–314) require more careful justification. For instance, regarding privacy preservation, the paper argues that Bank-Free T2I eliminates privacy concerns by internalizing semantic information into model parameters. However, traditional T2I models also encode semantic information within their parameters. Similarly, since standard T2I approaches do not require external memory banks, the claim of superior storage efficiency needs more substantiation.

3. To support the reported superiority in text-to-image (T2I) performance, it would be beneficial to include additional, widely recognized metrics such as T2I-CompBench[A] or HPSv2[B]. The current reliance on MJHQ-FID provides a narrow perspective on T2I performance. Incorporating these broader benchmarks, especially since the baseline PixArt-α also uses T2I-CompBench, would offer a more convincing evaluation.

4. The discussion on efficient downstream adaptation (Section 5.3) would benefit from deeper ablation studies and comparative analysis. While GMem demonstrates rapid adaptation in "absolute" training steps, efficiency should also be evaluated "relative" to strong baselines (e.g., REPA) by comparing computational overhead and convergence speed.

5. The manuscript would greatly benefit from extensive proofreading and consistency checks. For example, the important baseline PixArt-α is repeatedly miswritten as PixelArt-α, and the number of training epochs (450 in line 362) seems a wrong information. Although minor, such issues may confuse readers and reduce the paper’s overall polish and credibility.

[A] T2I-CompBench: A Comprehensive Benchmark for Open-world Compositional Text-to-image Generation, NeurIPS 2023

[B] Human Preference Score v2: A Solid Benchmark for Evaluating Human Preferences of Text-to-Image Synthesis

**Questions:**

None

---

> ### Author Response · Authors · 2025-11-25
> **Response to Reviewer Rz5N(1/3)**
>
> We sincerely thank the reviewer for the constructive and detailed feedback. We appreciate the acknowledgment of our method's conceptual soundness and experimental comprehensiveness.
>
> > W1: The dependency on REPA appears to be a crucial factor that requires further clarification.
>
> We sincerely thank the reviewer for the detailed review. We emphasize that GMem does not depend on REPA: it already brings substantial gains as a standalone module, while combining it with REPA yields additional, complementary improvements with amortized preprocessing overhead.
>
> | Method                | 12 epochs | 160 epochs |
> | --------------------- | --------- | ---------- |
> | LDiT-XL + REPA        | 19.97     | 1.84       |
> | LDiT-XL + GMem        | 13.95     | 1.58       |
> | LDiT-XL + REPA + GMem | 10.85     | 1.53       |
>
> **Empirical independence of GMem.** As shown in the table above, GMem already provides substantial gains as a standalone method: the GMem-only setting outperforms the REPA-only setting at both the early ($12$-epoch) and late ($160$-epoch) training stages on ImageNet $256\times 256$.
>
> **Rationale for combining GMem and REPA.** We justify the concurrent use of GMem and REPA based on their distinct acceleration mechanisms and engineering synergies:
>
> - **Orthogonal performance gains.** REPA acts as a training objective (modifying the loss via representation alignment), whereas GMem acts as an architectural enhancement (providing explicit semantic conditioning via memory snippets). They operate at orthogonal levels of the generative framework, so we argue that their benefits can be additive.
> - **Amortized preprocessing cost.** Both methods use a self-supervised vision encoder (e.g., DINOv2) during training. REPA uses the hidden states for loss calculation, while GMem uses the last-layer [CLS] token for memory construction. By integrating these two methods, vision encoder inference needs to be run only once per image, effectively amortizing the preprocessing cost of extracting memory snippets for GMem.

---

> ### Author Response · Authors · 2025-11-25
> **Response to Reviewer Rz5N(2/3)**
>
> > W2: **privacy preservation,** the paper argues that Bank-Free T2I eliminates privacy concerns by internalizing semantic information into model parameters. However, traditional T2I models also encode semantic information within their parameters.
>
> We apologize for the imprecise phrasing and thank the reviewer for this helpful comment. We clarify that our privacy and storage advantages are claimed **only relative to RAG-style T2I frameworks**, and we also emphasize that our bank-free GMem T2I pipeline remains highly data- and compute-efficient compared to traditional diffusion baselines.
>
> **Storage and privacy efficiency compared to retrieval-based models.** Retrieval-augmented diffusion models generally complement a base diffusion generator with an external database of precomputed features that is queried at training or inference time. For example, RDM requires on the order of tens of GBs (≈$32$ GB) of retrieval data and indices alongside the model weights. In contrast, both PixArt-$\alpha$ and our bank-free GMem T2I model do not maintain any persistent retrieval database or per-example feature bank.
>
> | Method                              | External DB (for retrieval)                           | Extra Storage (beyond model)              |
> | ----------------------------------- | ----------------------------------------------------- | ----------------------------------------- |
> | RDM (Retrieval-Augmented Diffusion) | CLIP embeddings for OpenImages/ArtBench + ScaNN index | ≈32 GB (≈11 GB embeddings + ≈21 GB index) |
> | PixArt-α                            | None                                                  | 0                                         |
> | GMem T2I (bank-free)            | None                                              | 0                                         |
>
> **Data and compute efficiency.** Independent of storage, our bank-free GMem T2I pipeline is highly data- and compute-efficient. Compared to PixArt-$\alpha$, GMem achieves a higher GenEval score ($0.52$ vs. $0.48$) while using only $0.20\text{M}$ paired text–image examples (vs. $25\text{M}$ for PixArt-$\alpha$) and about $78.5$ A100 GPU-days of total training compute (vs. $753$ A100 GPU-days for PixArt-$\alpha$). This is achieved by reusing a strong image-pretrained GMem backbone and fine-tuning only a small T2S module, instead of training a full T2I model from scratch.
>
> | Method              | Paired T2I Data (M pairs) | Total Compute (A100 GPU-days) | GenEval ↑ |
> | ------------------- | ------------------------- | ----------------------------- | --------- |
> | PixArt-α            | 25.0                      | 753                           | 0.48      |
> | GMem T2I (ours) | 0.20                  | 78.5                      | 0.52  |
>
> > W3: The current reliance on MJHQ-FID provides a narrow perspective on T2I performance.
>
> We thank the reviewer for the valuable suggestion. We agree that relying solely on FID can provide a limited view of T2I performance, which is why, as stated in Section 5.1, we utilize GenEval as our primary metric for Text-to-Image evaluation. We clarify MJHQ-FID is used only as a supplementary metric for specific style adaptation. GenEval rigorously validates compositional alignment by systematically evaluating compositional capabilities (e.g., object counts, attributes, spatial relations) using an object-focused evaluation set and it has been widely adopted as a primary benchmark in recent T2I evaluations (e.g., PixArt-$\alpha$, PixArt-$\Sigma$, and SANA).
>
> We summarize these significant gains across both Text-to-Image and Unified Multimodal tasks in the table below:
>
> | Task               | Method          | Data Usage     | Training Cost         | GenEval (↑) |
> | ------------------ | --------------- | -------------- | --------------------- | ----------- |
> | T2I Generation     | PixArt-$\alpha$ | 25M            | 753 GPU-days          | 0.48        |
> |                    | GMem (Ours)     | 1.48M ($1/17$) | 78.5 GPU-days ($1/9$) | 0.52        |
> | Unified Multimodal | OpenUni         | 23M            | 2.23 epochs           | 0.63        |
> |                    | +GMem (Ours)    | ~1.5M ($1/15$) | 0.07 epochs ($1/34$)  | 0.69        |
>
> On this benchmark, GMem demonstrates substantial advantages in both efficiency and quality compared to traditional approaches. As described in Sections 5.4 and H.1, GMem achieves competitive or superior GenEval scores while using about one-tenth as much data and computation.

---

> ### Author Response · Authors · 2025-11-25
> **Response to Reviewer Rz5N(3/3)**
>
> > W4: efficiency should also be evaluated "relative" to strong baselines (e.g., REPA) by comparing computational overhead and convergence speed
>
> We appreciate the reviewer’s valuable insight. We agree that a direct "relative" comparison against strong baselines such as REPA is essential, and we provide such an analysis below.
>
> **Conceptual advantage: GMem is an "open-set" generative model.** First, we highlight a fundamental structural difference: conventional class-conditional models (such as DiT and SiT) are essentially "closed" with respect to their label space—incorporating a new domain typically requires modifying the architecture (appending new class embeddings) and retraining them to map new labels to visual features. In contrast, GMem is "open": it conditions on semantic snippets not tied to a fixed vocabulary. This allows GMem to adapt to out-of-domain data immediately via the memory bank without architecture changes, offering a structural efficiency advantage that standard baselines lack.
>
> **Empirical relative comparison.** To rigorously quantify this advantage, we implemented a "relative" comparison on the FFHQ dataset using the LightningDiT-XL backbone. We compared GMem against two variants of REPA designed to handle label-space extension:
>
> - **REPA (new-class random init):** We append a new learnable class embedding $c_{\text{new}}$ initialized with a standard normal distribution. All FFHQ images are assigned to this label, and we fine-tune both the embedding and backbone.
> - **REPA (new-class avg-emb init):** To better evaluate knowledge transfer, we initialize $c_{\text{new}}$ as the average of all pre-trained class embeddings (excluding the null embedding). This allows the baseline to leverage the "average" semantic knowledge of the pre-trained model.
>
> Both GMem and the REPA baselines were fine-tuned under identical optimization settings. The results are shown in the table below:
>
> | Method              | Training speed | 0 (zero-shot) | 7 epochs | 180 Epochs |
> | ------------------- | -------------- | ------------- | -------- | ---------- |
> | GMem                | 1.55 s/step    | 8.65          | 8.55     | 7.08       |
> | REPA (random init)  | 1.62 s/step    | -             | 96.15    | 27.55      |
> | REPA (avg-emb init) | 1.62 s/step    | -             | 74.28    | 21.32      |
>
> **Benefit of initialization.** Comparing the two REPA variants, REPA (avg-emb init) indeed accelerates convergence (FID $74.28$ vs $96.15$ at $7$ epochs), confirming that utilizing prior knowledge helps.
>
> **Superiority of GMem.** However, GMem radically outperforms even REPA (avg-emb init).
>
> - **Zero-shot adaptation.** GMem achieves a strong FID of $8.65$ immediately without any gradient updates, whereas the REPA baselines cannot generate valid samples for the new class in the zero-shot setting.
> - **Adaptation speed.** With only $7$ epochs, GMem maintains high image quality (FID $8.55$), whereas REPA (avg-emb init) yields poor results (FID $74.28$). This substantial performance advantage persists even after prolonged training ($180$ epochs).
>
> We argue that these results empirically validate GMem as a general-purpose base model tailored for rapid downstream adaptation, substantiating one of the primary contributions of this work.
>
> > W5 – The manuscript would greatly benefit from extensive proofreading and consistency checks.
>
> We sincerely thank the reviewer for the meticulous reading and for pointing out these errors. We have carefully checked the manuscript and corrected them in the revised version:
>
> - **Typo correction (PixArt-$\alpha$).** We apologize for the repeated misspelling of "PixArt-$\alpha$" as "PixelArt-$\alpha$". We have globally corrected this proper noun throughout the manuscript (e.g., in Figure 1, Table 2, Table 15, and Section 5.4) to ensure consistency.
> - **Factual correction (Line 362).** "$450$ epochs" in line 362 is a typo, and the correct number should be $52$ epochs as illustrated in Table 1.
>
> We will also conduct a thorough proofreading of the entire paper to eliminate any remaining grammatical errors or inconsistencies.

---

### Author Response · Authors · 2025-12-03
**General Response**

We sincerely thank the reviewers for their thoughtful feedback. We are encouraged that

- the **challenge** we address was characterized as timely and important (Reviewers Rz5N, 7j2m);
- our **presentation** was seen as well-structured and easy to follow (All Reviewers Rz5N, gjUM, 7j2m, 82a5);
- our **idea** was viewed as conceptually fresh, sound, and thoughtfully motivated (Reviewers 7j2m, Rz5N), and practical to reproduce elsewhere (Reviewer 82a5);
- our **experiments** were recognized as comprehensive and broad (Reviewers Rz5N, gjUM, 82a5);
- the **improvements** were judged significant—with order-of-magnitude speedups and strong gains over baselines (All Reviewers Rz5N, gjUM, 7j2m, 82a5);
- and the method shows **broad applicability** with rapid downstream adaptation and transfer learning potential (Reviewers 7j2m, 82a5).

Furthermore, our contributions can be divided into three crucial parts:

- introducing a memory-augmented diffusion framework (GMem) that accelerates training by a large margin—up to an order-of-magnitude speedup (Reviewers Rz5N, 7j2m, 82a5);
- enabling rapid downstream adaptation, underscoring broad applicability to new domains (Reviewers 7j2m, 82a5);
- unifying image-only and text-to-image diffusion modeling into a data- and compute-efficient T2I pathway (Reviewers gjUM, Rz5N).

### Summary of Contribution and Novelty

Our work stands out through the following key contributions and innovations:

1. **Novel perspective on conditional diffusion.**
   - **Conditioning view of conditional diffusion.** We frame conditional diffusion along a *conditioning-strength continuum*—linking class-conditional (weak), GMem via image-semantic snippets (medium), and T2I (strong)—which was described as *conceptually fresh and thoughtfully motivated* and *well-structured* and *easy to follow* by Reviewers 7j2m and 82a5.
   - **Explicit semantic memory for controllable conditioning.** We introduce memory snippets as compact, controllable semantic conditioning; manipulating them lets us dial conditioning strength with the architecture unchanged, which was recognized as conceptually sound and practical to reproduce with clear storage/complexity considerations by Reviewers Rz5N and 82a5.
   - **Unified weak→strong pathway (image-only → T2I).** This view yields a single route that progressively strengthens conditioning—pretraining with snippets and transitioning to text latents—*avoiding separate models*; this unified, data- and compute-efficient T2I pathway was acknowledged by Reviewers gjUM and Rz5N.
2. **Well-designed experimental validation (comprehensive and broad).**
   - **Datasets & scales.** We evaluate across datasets and scales—from CIFAR-10 to ImageNet-1K at 256×256/512×512, plus T2I—a breadth recognized as comprehensive and broad by Reviewers Rz5N and gjUM, with generalizability across encoders noted by Reviewer 82a5.
   - **Metrics.** We report metrics covering quality (FID on CIFAR-10/ImageNet; GenEval and MJHQ-FID for T2I), efficiency (Epochs, wall-clock time, NFE), and diversity (both quantitative and qualitative comparisons), acknowledged by Reviewers gjUM and 82a5.
   - **Tasks.** We evaluate class-conditional image generation, text-to-image synthesis, downstream domain adaptation to new visual domains (e.g., anime/faces) from an ImageNet-pretrained GMem, and unified generation within a single framework (image-only → T2I)—recognized by Reviewers gjUM and 7j2m.
3. **Strong improvements on three critical axes.**
   - **Training efficiency.** Conditioning on semantically rich snippets simplifies learning dynamics and delivers order-of-magnitude speedups while maintaining or improving quality—recognized by Reviewers 7j2m, 82a5, and gjUM.
   - **Adaptability.** Snippet-based conditioning is not tied to a fixed vocabulary, enabling rapid downstream adaptation to new domains without architectural changes—recognized by Reviewers 7j2m and 82a5.
   - **Data/compute efficiency for T2I.** The unified pathway reuses image-pretrained GMem for T2I, achieving data- and time-efficient training while remaining competitive—recognized by Reviewers Rz5N and gjUM.
4. **Potential applications.**
   - This paper examines diffusion modeling through the lens of conditioning, offering a unified perspective that connects class-conditional and T2I generation, and introduces GMem, which unifies these two frameworks; this perspective was noted as conceptually fresh and thoughtfully motivated by Reviewer 7j2m.
   - The approach could potentially enable transfer learning—rapid adaptation to new domains with minimal effort—as recognized by Reviewers 7j2m and 82a5.
   - It sheds light on training-efficient conditional generation (order-of-magnitude speedups while maintaining quality) and a data-/compute-efficient T2I pathway via unified conditioning—efficiency recognized by Reviewers 7j2m, 82a5, and gjUM.

---

### Meta-Review · Area_Chair_aLFa · 2026-01-07

**Summary:**

This paper proposes GMem, a memory-augmented diffusion framework aimed at accelerating training and enabling efficient adaptation.

While the idea is interesting and the problem is relevant, reviewers raised concerns about the conceptual novelty, strength of empirical evidence, and fairness of efficiency comparisons, which ultimately limit the impact of the work.

Although the method shows promising speedups, the overall contribution was not found to be sufficiently clear or convincing for acceptance.

**Reviewer Concerns:**

- Reviewer 7j2m and Reviewer 82a5 found the rebuttal helpful and agreed that several clarification and implementation issues were addressed, particularly regarding adaptation and presentation.

- Novelty remains limited, with concerns that GMem is conceptually close to existing representation-augmented or retrieval-based diffusion methods.

- Dependence on strong baselines (e.g., REPA) is not fully resolved; it remains unclear how much of the reported gains come uniquely from GMem.

- Efficiency claims rely heavily on selected settings and do not consistently demonstrate advantages under fair, relative comparisons.

- Evaluation scope, especially for text-to-image generation, is still considered incomplete, relying on a limited set of metrics.

- Some reviewers remain unconvinced that the proposed memory mechanism provides a clear conceptual advance beyond engineering improvements.

**Reviewer Scores:**

+ Reviewer 7j2m increases score 4-->6
+ Reviewer 82a5 maintains score 8
- Other reviewers keep their original scores with reject (4)

---

### Decision · Program_Chairs · 2026-01-26

Reject